

# Benchmarking a Catchment-Aware Long Short-Term Memory Network (LSTM) for Large-Scale Hydrological Modeling

Frederik Kratzert[1], Daniel Klotz[1], Guy Shalev[2], Günter Klambauer[1], Sepp Hochreiter[1,*], and
Grey Nearing[3,*]

[1]LIT AI Lab & Institute for Machine Learning, Johannes Kepler University Linz, Austria
[2]Google Research
[3]Department of Geological Sciences, University of Alabama, Tuscaloosa, AL United States
[*]These authors contributed equally to this work.

**Correspondence:** Frederik Kratzert (kratzert@ml.jku.at)

**Abstract.** Regional rainfall-runoff modeling is an old but still mostly out-standing problem in Hydrological Sciences. The problem currently is that traditional hydrological models degrade significantly in performance when calibrated for multiple basins together instead of for a single basin alone. In this paper, we propose a novel, data-driven approach using Long Short-Term Memory networks (LSTMs), and demonstrate that under a 'big data' paradigm, this is not necessarily the case. By training
a single LSTM model on 531 basins from the CAMELS data set using meteorological time series data and static catchment attributes, we were able to significantly improve performance compared to a set of several different hydrological benchmark models. Our proposed approach not only significantly outperforms hydrological models that were calibrated regionally but also achieves better performance than hydrological models that were calibrated for each basin individually. Furthermore, we propose an adaption to the standard LSTM architecture, which we call an Entity-Aware-LSTM (EA-LSTM), that allows for learning,
and embedding as a feature layer in a deep learning model, catchment similarities. We show that this learned catchment similarity corresponds well with what we would expect from prior hydrological understanding.

## 1 Introduction

A longstanding problem in the Hydrological Sciences is about how to use one model, or one set of models, to provide spatially continuous hydrological simulations across large areas (e.g., regional, continental, global). This is the so-called *regional*
*modeling problem*, and the central challenge is about how to extrapolate hydrologic information from one area to another – e.g., from gauged to ungauged watersheds, from instrumented to non-instrumented hillslopes, from areas with flux towers to areas without, etc. (Blöschl and Sivapalan, 1995). Often this is done using ancillary data (e.g. soil maps, remote sensing, digital elevation maps, etc.) to help understand similarities and differences between different areas. The regional modeling problem is thus closely related to the problem of prediction in ungauged basins (Blöschl et al., 2013; Sivapalan et al., 2003). This problem
is well-documented in several review papers, therefore we point the interested reader to the comprehensive reviews by Razavi and Coulibaly (2013) and Hrachowitz et al. (2013), and to the more recent review in the introduction by Prieto et al. (2019).



Currently, the most successful hydrological models are calibrated to one specific basin, whereas a regional model must be somehow 'aware' of differences between hydrologic behaviors in different catchments (e.g., ecology, geology, pedology, topography, geometry, etc.). The challenge of regional modeling is to learn and encode these differences so that differences in

catchment characteristics translate into appropriately heterogeneous hydrologic behavior. Razavi and Coulibaly (2013) recognize two primary types of strategies for regional modeling: *model-dependent* methods and *model-independent* (data-driven) methods. Here, model-dependent denotes approaches where regionalization explicitly depends on a pre-defined hydrological model (e.g., classical process-based models), while model-independent denotes data-driven approaches that do not include a specific model. The critical difference is that the first tries to derive hydrologic parameters that can be used to run simulation

models from available data (i.e., observable catchment characteristics). In this case, the central challenge is the fact that there is typically strong interaction between individual model parameters (e.g., between soil porosity and soil depth, or between saturated conductivity and an infiltration rate parameter), such that any meaningful joint probability distribution over model parameters will be complex and multi-modal. This is closely related to the problem of equifinality (Beven and Freer, 2001).

Model-dependent regionalization has enjoyed major attention from the hydrological community, so that today a large va-

riety of approaches exist. To give a few selective examples, Seibert (1999) calibrated a conceptual model for 11 catchments and regressed them against the available catchment characteristics. The regionalization capacity was tested against seven other catchments, where the reported performance ranged between an Nash-Sutcliffe Efficiency (NSE) of 0.42 and 0.76. Samaniego et al. (2010) proposed a multiscale parameter regionalization (MPR) method, which simultaneously sets up the model and a regionalization scheme by regressing the global parameters of a set of a-priori defined transfer functions that map from an-

cillary data like soil properties to hydrological model parameters. Beck et al. (2016) calibrated a conceptual model for 1787 catchments around the globe and used these as a catalog of 'donor catchments', and then extend this library to new catchments by identifying the ten most similar catchments from the library in terms of climatic and physiographic characteristics to parameterize a simulation ensemble. Prieto et al. (2019) first regionalized hydrologic signatures (Gupta et al., 2008) using a regression model (random forests), and then calibrated a rainfall runoff model to the regionalized hydrologic signatures.

Model-independent methods, in contrast, do not rely on prior knowledge of the hydrological system. Instead, these methods learn the entire mapping from ancillary data and meteorological inputs to streamflow or other output fluxes directly. A model of this type has to 'learn' how catchment attributes or other ancillary data distinguish different catchment response behaviours. However, hydrological modeling typically provides the most accurate predictions when a model is calibrated to a single specific catchment (Mizukami et al., 2017), whereas data-driven approaches might benefit from large cross-section of diverse training

data, because knowledge can be transferred across sites. Among the category of data-driven approaches are neural networks. Besaw et al. (2010) showed that an artificial neural network trained on one catchment (using only meteorological inputs) could be moved to a similar catchment (during a similar time period). However, the accuracy of their network in the *training* catchment was only a NSE of 0.29. Recently, Kratzert et al. (2018b) have shown, that Long Short-Term Memory (LSTM) networks, a special type of recurrent neural networks, are well suited for the task of rainfall-runoff modeling. This study already included

first experiments towards regional modeling, while still using only meteorological inputs and ignoring ancillary catchment attributes. In a preliminary study Kratzert et al. (2018c) demonstrated that their LSTM-based approach outperforms, on av-





erage, a well-calibrated Sacramento Soil Moisture Accounting Model (SAC-SMA) in an asymmetrical comparison where the LSTM was used in an *ungauged* setting and SAC-SMA was used in a *gauged* setting - i.e., SAC-SMA was calibrated individually for each basin whereas the LSTM never saw training data from any catchment where it was used for prediction. This
was done by providing the LSTM-based model with meteorological forcing-data and additional catchment attributes. From these preliminary results we can already assume that this general modeling approach is promising and has the potential for regionalization.

The objectives of this study are:

(i) to demonstrate that we can use large-sample hydrology data (Gupta et al., 2014; Peters-Lidard et al., 2017) to develop a
regional rainfall-runoff model that capitalizes on observable ancillary data in the form of catchment attributes to produce accurate streamflow estimates over a large number of basins,

(ii) to benchmark the performance of our neural network model against several existing hydrology models, and

(iii) to show how the model uses information about catchment characteristics to differentiate between different rainfall-runoff behaviors.

To this end we built an LSTM-based model that learns catchment similarities directly from meteorological forcing-data and ancillary data of multiple basins and evaluate its performance in a 'gauged' setting, meaning that we never ask our model to predict in a basin where it did not see training data. Concretely, we propose an adaption of the LSTM where catchment attributes explicitly control which parts of the network are used for a given basin. Because the model is trained using both catchment attributes and meteorological time series data, to predict streamflow, it can learn how to combine different parts
of the network to simulate different types of rainfall-runoff behaviors. In principle, the approach explicitly allows for sharing parts of the networks for similarly behaving basins, while using different independent parts for basins with completely different rainfall-runoff behavior. Furthermore, our adaption provides a mapping from catchment attribute space into a learned, high-dimensional space, i.e. a so-called embedding, in which catchments with similar rainfall-runoff behavior can be placed together. This embedding can be used to preform data-driven catchment similarity analysis.

The paper is organized as follows. Section 2 (Methods) describes our LSTM-based model, the data, the benchmark hydrological models, and the experimental design. Section 3 (Results) presents our modelling results, the benchmarking results and the results of our embedding layer analysis. Section 4 (Discussion and Conclusion) reviews certain implications of our model and results, and summarizes the advantages of using data-driven methods for extracting information from catchment observables for regional modeling.





## 2 Methods

### 2.1 A Brief Overview of the Long Short-Term Memory network

An LSTM network is a type of recurrent neural network that includes dedicated memory cells that store information over long time periods. A specific configuration of operations in this network, so-called gates, control the information flow within the LSTM (Hochreiter and Schmidhuber, 1997). These memory cells are, in a sense, analogous to a state vector in a traditional dynamical systems model, which makes LSTMs potentially an ideal candidate for modeling dynamical systems like watersheds. Compared to other types of recurrent neural networks, LSTMs do not have a problem with exploding and/or vanishing gradients, which allows them to learn long-term dependencies between input and output features. This is desirable for modeling catchment processes like snow-accumulation and snow-melt that have relatively long time scales compared with the timescales of purely input driven domains (i.e., precipitation events).

An LSTM works as follows (see also Fig. 1a): Given an input sequence $\boldsymbol{x} = [\boldsymbol{x}[1], .., \boldsymbol{x}[T]]$ with $T$ time steps, where each element $\boldsymbol{x}[t]$ is a vector containing input features (model inputs) at time step $t$ ($1 \leq t \leq T$), the following equations describe the forward pass through the LSTM:

$$\boldsymbol{i}[t] = \sigma(\boldsymbol{W_i}\boldsymbol{x}[t] + \boldsymbol{U_i}\boldsymbol{h}[t-1] + \boldsymbol{b_i}) \tag{1}$$

$$\boldsymbol{f}[t] = \sigma(\boldsymbol{W_f}\boldsymbol{x}[t] + \boldsymbol{U_f}\boldsymbol{h}[t-1] + \boldsymbol{b_f}) \tag{2}$$

$$\boldsymbol{g}[t] = \tanh(\boldsymbol{W_g}\boldsymbol{x}[t] + \boldsymbol{U_g}\boldsymbol{h}[t-1] + \boldsymbol{b_g}) \tag{3}$$

$$\boldsymbol{o}[t] = \sigma(\boldsymbol{W_o}\boldsymbol{x}[t] + \boldsymbol{U_o}\boldsymbol{h}[t-1] + \boldsymbol{b_o}) \tag{4}$$

$$\boldsymbol{c}[t] = \boldsymbol{f}[t] \odot \boldsymbol{c}[t-1] + \boldsymbol{i}[t] \odot \boldsymbol{g}[t] \tag{5}$$

$$\boldsymbol{h}[t] = \boldsymbol{o}[t] \odot \tanh(\boldsymbol{c}[t]), \tag{6}$$

where $\boldsymbol{i}[t]$, $\boldsymbol{f}[t]$ and $\boldsymbol{o}[t]$ are the *input gate*, *forget gate*, and *output gate*, respectively, $\boldsymbol{g}[t]$ is the *cell input* and $\boldsymbol{x}[t]$ is the *network input* at time step $t$ ($1 \leq t \leq T$), $\boldsymbol{h}[t-1]$ is the *recurrent input* $\boldsymbol{c}[t-1]$ the *cell state* from the previous time step. At the first time step, the hidden and cell states are initialized as a vector of zeros. $\boldsymbol{W}$, $\boldsymbol{U}$ and $\boldsymbol{b}$ are learnable parameters for each gate, where subscripts indicate which gate the particular weight matrix/vector is used for, $\sigma(\cdot)$ is the sigmoid-function, $\tanh(\cdot)$ the hyperbolic tangent function and $\odot$ is element-wise multiplication. The intuition behind this network is that the cell states ($\boldsymbol{c}[t]$) characterize the memory of the system. The cell states can get modified by the forget gate ($\boldsymbol{f}[t]$), which can delete states, and the input gate ($\boldsymbol{i}[t]$) and cell update ($\boldsymbol{g}[t]$), which can add new information. In the latter case, the cell update is seen as the information that is added and the input gate controls into which cells new information is added. Finally, the output gate ($\boldsymbol{o}[t]$) controls which information, stored in the cell states, is outputted. For a more detailed description, as well as a hydrological interpretation, see Kratzert et al. (2018b).





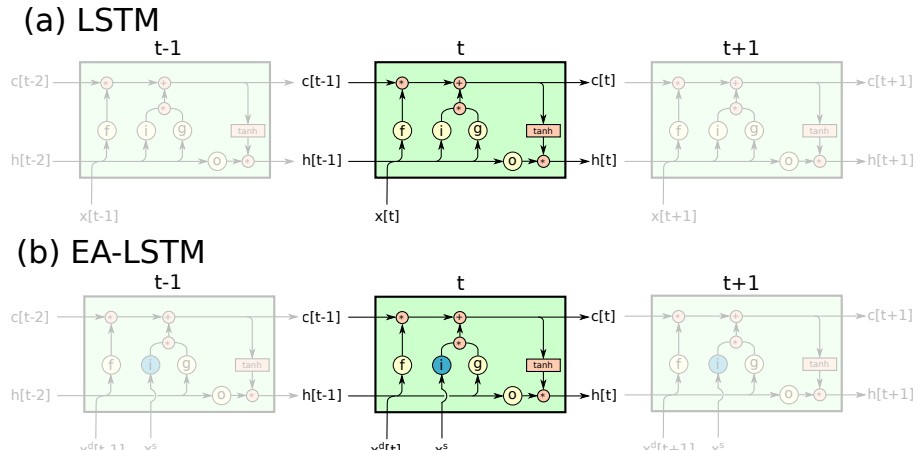

**Figure 1.** Visualization of (a) the standard LSTM cell as defined by Eq. (1-6) and (b) the proposed Entity-Aware-LSTM (EA-LSTM) cell as defined by Eq. (7-12).

## 2.2 A New Type of Recurrent Network: The Entity-Aware-LSTM

To reiterate from the introduction, our objective is to build a network that learns catchment similarities directly from rainfall-runoff data in multiple basins. To achieve this, it is necessary to provide the network with information on the catchment characteristics that contain some amount of information that allows for discriminating between different catchments. Ideally, we want the network to condition the *processing of the dynamic inputs* on a set of static catchment characteristics. That is, we want the network to learn a mapping from meteorological time series into streamflow that itself (i.e., the mapping) depends on
a set of static catchment characteristics that could, in principle, be measured anywhere in our modeling domain.

One way to do this would be to add the static features as additional inputs at every time step. That is, we could simply augment the vectors $x[t]$ at every time step with a set of catchment characteristics that do not (necessarily) change over time. However, this approach does not allow us to directly inspect what the LSTM learns from these static catchment attributes.

Our proposal is therefore to use a slight variation on the normal LSTM architecture (an illustration is given in Fig. 1b):

$$i = \sigma(W_i x_s + b_i) \tag{7}$$

$$f[t] = \sigma(W_f x_d[t] + U_f h[t-1] + b_f) \tag{8}$$

$$g[t] = \tanh(W_g x_d[t] + U_g h[t-1] + b_g) \tag{9}$$

$$o[t] = \sigma(W_o x_d[t] + U_o h[t-1] + b_o) \tag{10}$$

$$c[t] = f[t] \odot c[t-1] + i \odot g[t] \tag{11}$$

$$h[t] = o[t] \odot \tanh(c[t]) \tag{12}$$

Here $i$ is an input gate, which now does not change over time. $x_s$ are the static inputs (e.g., catchment attributes) and $x_d[t]$ are the dynamic inputs (e.g., meteorological forcings) at time step $t$ ($1 \leq t \leq T$). The rest of the LSTM remains unchanged. The





intuition is as follows: we explicitly process the static inputs $x_s$ and the dynamic inputs $x_d[t]$ separately within the architecture and assign them special tasks. The static features control, through input gate ($i$), which parts of the LSTM are activated for

any individual catchment, while the dynamic inputs control what information is written into the memory ($g[t]$), what is deleted ($f[t]$), and what of the stored information to output ($o[t]$) at the current time step $t$.

We are calling this an **Entity-Aware-LSTM (EA-LSTM)** because it explicitly differentiates between similar types of dynamical behaviors (here rainfall-runoff processes) that differ between individual entities (here different watersheds). After training, the static input gate of the **EA-LSTM** contains a series of real values in the range (0,1) that allow certain parts of

the input gate to be active through the simulation of any individual catchment. In principle, different groups of catchments can share different parts of the full trained network.

This is an embedding layer, which allows for a non-naive information sharing between the catchments. For example, we could potentially discover, after training, that two particular catchments share certain parts of the activated network based on geological similarities while other parts of the network remain distinct due to ecological dissimilarities. This embedding layer

allows for complex interactions between catchment characteristics, and - importantly - makes it possible for those interactions to be directly informed by the rainfall-runoff data from all catchments used for training.

## 2.3    Objective Function: A Smooth Joint NSE

An objective function is required for training the network. For regression tasks such as runoff prediction, the mean-squared-error (MSE) is commonly used. Hydrologists also sometimes use the NSE because it has an interpretable range of ($-\infty$, 1).

Both the MSE and NSE are squared error loss functions, with the difference being that the latter is normalized by the total variance of the observations. For single-basin optimization, the MSE and NSE will typically yield the same optimum parameter values, discounting any effects in the numerical optimizer that depend on the absolute magnitude of the loss value.

The linear relation between these two metrics (MSE and NSE) is lost, however, when calculated over data from multiple basins. In this case, the means and variances of the observation data are no longer constant because they differ between basins.

We will exploit this fact. In our case, the MSE from a basin with low average discharge (e.g. smaller, arid basins) is generally smaller than the MSE from a basin with high average discharge (e.g. larger, humid basins). We need an objective function that does not depend on basin-specific mean discharge so that we do not overweight large humid basins (and thus perform poorly on small, arid basins). Our loss function is therefore the average of the NSE values calculated at each basin that supplies training data – referred to as basin averaged Nash-Sutcliffe Efficiency (NSE*). Additionally, we add a constant term to the denominator

($\epsilon = 0.1$), the variance of the observations, so that our loss function does not explode (to negative infinity) for catchments with very low flow-variance. Our loss function is therefore:

$$\text{NSE*} = \frac{1}{B}\sum_{b=1}^{B}\sum_{n=1}^{N}\frac{(\widehat{y}_n - y_n)^2}{(s(b)+\epsilon)^2}, \tag{13}$$

where $B$ is the number of basins, $N$ is the number of samples (days) per basin $B$, $\widehat{y}_n$ is the prediction of sample $n$ ($1 \leq n \leq N$), $y_n$ the observation and $s(b)$ is the standard deviation of the discharge in basin $b$ ($1 \leq b \leq B$), calculated from the training





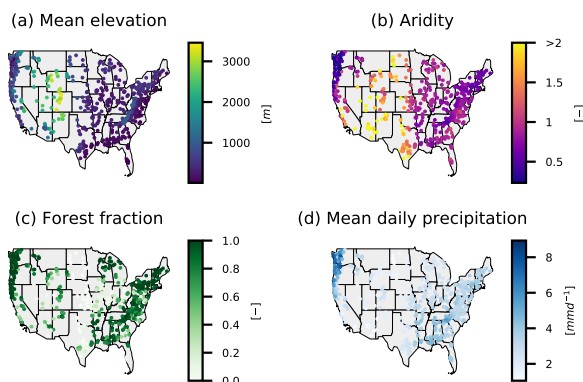

**Figure 2.** Overview of the basin location and corresponding catchment attributes. (a) The mean catchment elevation, (b) the catchment aridity (PET/P), (c) the fraction of the catchment covered by forest and (d) the daily average precipitation

period. In general, an entity-aware deep learning model will need a loss function that does not underweight entities with lower (relative to other entities in the training data set) absolute values in the target data.

## 2.4 The NCAR CAMELS Dataset

To benchmark our proposed EA-LSTM model, and to assess its ability to learn meaningful catchment similarities, we will use the Catchment Attributes and Meteorological (CAMELS) data set (Newman et al., 2014; Addor et al., 2017b). CAMELS is a
set of data concerning 671 basins that is curated by the US National Center for Atmospheric Research (NCAR). The CAMELS basins range in size between 4 and 25 000 km$^2$, and were chosen because they have relatively low anthropogenic impacts. These catchments span a range of geologies and ecoclimatologies, as described in Newman et al. (2015) and Addor et al. (2017a).

We used the same 531 basins from the CAMELS data set as Newman et al. (2017). The basins are mapped in Fig. 2. These
basins were chosen out of the full set because some of the basins have a large ($> 10$ %) discrepancy between different strategies for calculating the basin area, and incorrect basin area would introduce significant uncertainty into a modeling study. The basin selection and subset is described by Newman et al. (2017).

For time-dependent meteorological inputs ($x_d[t]$), we used the daily, basin-averaged Maurer forcings (Wood et al., 2002) supplied with CAMELS. Our input data includes: (i) daily cumulative precipitation, (ii) daily minimum air temperature, (iii)
daily maximum air temperature, (iv) average short-wave radiation and (v) vapor pressure. Furthermore, 27 CAMELS catchment characteristics were used as static input features ($x_s$); these were chosen as a subset of the full set of characteristics explored by Addor et al. (2017b) that are derivable from remote sensing or CONUS-wide available data products. These catchment attributes include climatic and vegetation indices, as well as soil and topographical properties (see Tab. A1 for an exhaustive list).



## 2.5 Benchmark models

The first part of this study benchmarks our proposed model against several high-quality benchmarks. The purpose of this exercise is to show that the EA-LSTM provides reasonable hydrological simulations.

To do this, we collected a set of existing hydrological models[1] that were configures, calibrated, and run by several previous studies over the CAMELS catchments. These models are: (i) SAC-SMA (Burnash et al., 1973; Burnash, 1995) coupled with the Snow-17 snow routine (Anderson, 1973), hereafter referred to as SAC-SMA, (ii) VIC (Liang et al., 1994), (iii) FUSE (Clark et al., 2008; Henn et al., 2008) (three different model structures, 900, 902, 904), (iv) HBV (Seibert and Vis, 2012) and (v) mHM (Samaniego et al., 2010; Kumar et al., 2013). In some cases, these models were calibrated to individual basins, and in other cases they were not. All of these benchmark models were run by other groups - we did not run any of our own benchmarks. We chose to use existing model runs so to not bias the calibration of the benchmarks to possibly favor of our own model. Each set of simulations that we used for benchmarking is documented elsewhere in the Hydrology literature (references below). Each of these benchmark models use the same daily Maurer forcings that we used with our EA-LSTM, and all were calibrated and validated on the same time period(s). These benchmark models can be distinguished into two different groups:

1. **Models calibrated for each basin individually**: These are SAC-SMA (Newman et al., 2017), VIC (Newman et al., 2017), FUSE[2], mHM (Mizukami et al., 2019) and HBV (Seibert et al., 2018). The HBV model supplied both a lower and upper benchmark, where the lower benchmark is an ensemble mean of 1000 uncalibrated HBV models and the upper benchmark is an ensemble of 100 calibrated HBV models.

2. **Models that were regionally calibrated**: These share one parameter set for all basins in the data set. Here we have calibrations of the VIC model (Mizukami et al., 2017) and mHM (Rakovec et al., 2019).

## 2.6 Experimental Setup

All model calibration and training was performed using data from the time period 1 October 1999 through 30 September 2008. All model and benchmark evaluation was done using data from the time period 1 Oct 1989 through 30 September 1999. We trained a single LSTM or EA-LSTM model using calibration period data from all basins, and evaluated this model using validation period data from all basins. This implies that a single parameter set (i.e. W, U, b from Eq. 1-4 and Eq. 7-10) was trained to work across all basins.

We trained and tested the following three model configurations:

– **LSTM without static inputs**: A single LSTM trained on the combined calibration data from all basins, using only the meteorological forcing data and ignoring static catchment attributes.

---

[1]Will be released on HydroShare. DOI will be added for publication.

[2]The FUSE runs were generated by Nans Addor (n.addor@uea.ac.uk) and given to us by personal communication. These runs are part of current development by N. Addor on the FUSE model itself and might not reflect the final performance of the FUSE model.





- **LSTM with static inputs**: A single LSTM trained on the combined calibration data of all basins, using the mete-orological features as well as the static catchment attributes. These catchment descriptors were concatenated to the meteorological inputs at each time step.

- **EA-LSTM with static inputs**: A single EA-LSTM trained on the combined calibration data of all basins, using the meteorological features as well as the static catchment attributes. The catchment attributes were input to the static input gate in Eq. 7, while the meteorological inputs were used at all remaining parts of the network (Eq. 8-10).

All three model configurations were trained using the squared-error performance metrics discussed in Sect. 2.3 (MSE and NSE*). This resulted in six different model/training configurations.

To account for stochasticity in the network initialization and in the optimization procedure (we used stochastic gradient descent), all networks were trained with $n = 8$ different random seeds. Predictions from the different seeds were combined into an ensemble by taking the mean prediction at each timestep of all $n$ different models under each configuration. In total, we trained and tested six different settings and eight different models per setting for a total of 48 different trained LSTM-type models. For all LSTMs we used the same architecture (apart from the inclusion of a static input gate in the EA-LSTM), which we found through hyperparameter optimization (see Appendix B for more details about the hyperparameter search). The LSTMs had 256 memory cells and a single fully connected layer with a dropout rate (Srivastava et al., 2014) of 0.4. The LSTMs were run in sequence-to-value mode (as opposed to sequence-to-sequence mode), so that to predict a single (daily) discharge value required meteorological forcings from 269 preceding days, as well as the forcing data of the target day, making the input sequences 270 time steps long.

### 2.6.1 Assessing Model Performance

Because no one evaluation metric can fully capture the consistency, reliability, accuracy, and precision of a streamflow model, it was necessary to use a variety of performance metrics for model benchmarking (Gupta et al., 1998). Evaluation metrics used to compare models are listed in Tab. 1. These metrics focus specifically on assessing the ability of the model to capture high-flows and low-flows, as well as assessing overall performance using a decomposition of the standard squared error metrics that is less sensitive to bias (Gupta et al., 2009).

### 2.6.2 Robustness and Feature Ranking

All catchment attributes used in this study are derived from gridded data products (Addor et al., 2017a). Taking the catchment's mean elevation as an example, we would get different mean elevations depending on the resolution of the gridded digital elevation model. More generally, there is uncertainty in all CAMELS catchment attributes. Thus, it is important that we evaluate the robustness of our model and of our embedding layer (particular values of the 256 static input gates) to changes in the exact values of the catchment attributes. Additionally, we want some idea about the relative importance of different catchment attributes.





**Table 1.** Overview of used evaluation metrics. The notation of the original publications is kept.

| Metric | Reference | Equation |
|---|---|---|
| Nash-Sutcliff-Efficiency (NSE) | Nash and Sutcliffe (1970) | $1 - \frac{\sum_{t=1}^{T}(Q_m[t]-Q_o[t])^2}{\sum_{t=1}^{T}(Q_o[t]-\bar{Q_o})^2}$ |
| $\alpha$-NSE Decomposition | Gupta et al. (2009) | $\sigma_s/\sigma_o$ |
| $\beta$-NSE Decomposition | Gupta et al. (2009) | $(\mu_s - \mu_o)/\sigma_o$ |
| Top 2% peak flow bias (FHV) | Yilmaz et al. (2008) | $\frac{\sum_{h=1}^{H}(QS_h-QO_h)}{\sum_{h=1}^{H}QO_h} \times 100$ |
| Bias of FDC midsegment slope (FMS) | Yilmaz et al. (2008) | $\frac{(\log(QS_{m1})-\log(QS_{m2}))-(\log(QO_{m1})-\log(QO_{m2}))}{(\log(QO_{m1})-\log(QO_{m2}))} \times 100$ |
| 30% low flow bias (FLV) | Yilmaz et al. (2008) | $\frac{\sum_{l=1}^{L}(\log(QS_l)-\log(QS_L))-\sum_{l=1}^{L}(\log(QO_l)-\log(QO_L))}{\sum_{l=1}^{L}(\log(QO_l)-\log(QO_L))} \times 100$ |

To estimate the robustness of the trained model to uncertainty in the catchment attributes, we added Gaussian noise $\mathcal{N}(0,\sigma)$
with increasing standard deviation to the individual attribute values and assessed resulting changes in model performance
for each noise level. Concretely, additive noise was drawn from normal distributions with 10 different standard deviations:
$\sigma = [0.1, 0.2, \ldots, 0.9, 1.0]$. All input features (both static and dynamic) were standardized (zero mean, unit variance) before
training, so these perturbation sigmas did not depend on the units or relative magnitudes of the individual catchment attributes.
For each basin and each standard deviation we drew 50 random noise vectors, resulting in $531 * 10 * 50 = 265500$ evaluations
of each trained EA-LSTM.

To provide a simple estimate of the most important static features, we used the method of Morris (Morris, 1991). Albeit
the Morris method is relatively simple, it has been shown to provide meaningful estimations of the global sensitivity and is
widely used (e.g., Herman et al., 2013; Wang and Solomatine, 2019). The method of Morris uses an approximation of local
derivatives, which can be extracted directly from neural networks without additional computations, which makes this a highly
efficient method of sensitivity analysis.

The method of Morris typically estimates feature sensitivities ($\text{EE}_i$) from local (numerical) derivatives.

$$\text{EE}_i = \frac{f(x_1, ..., x_i + \triangle_i, ..., x_p) - f(x)}{\triangle_i}, \tag{14}$$

Neural networks are completely differentiable (to allow for back-propagation) and thus it is possible to calculate the exact
gradient with respect to the static input features. Thus, for neural networks the method of Morris can be applied analytically.

$$\text{EE}_i = \lim_{\triangle_i \to 0} \frac{f(x_1, ..., x_i + \triangle_i, ..., x_p) - f(x)}{\triangle_i} = \frac{\partial f(x)}{\partial x_i}, \tag{15}$$

This makes it unnecessary to run computationally expensive sampling methods to approximate the local gradient. Further,
since we predict one time step of discharge at the time, we obtain this sensitivity measure for each static input for each day
in the validation period. A global sensitivity measure for each basin and each feature is then derived from taking the average
absolute gradient (Saltelli et al., 2004).





### 2.6.3 Analysis of Catchment Similarity from the Embedding Layer

Once the model is trained, the input gate vector ($i$, see Eq. 7) for each catchment is fixed for the simulation period. This results in a vector that represents an embedding of the static catchment features (here in $\mathbb{R}^{27}$) into the high-dimensional space of the LSTM (here in $\mathbb{R}^{256}$). The result is a set of real-valued numbers that map the catchment characteristics onto a strength, or weight, associated with each particular cell state in the EA-LSTM. This weight controls how much of the cell input ($g[t]$, see Eq. 9) is written into the corresponding cell state ($c[t]$, see Eq. 11).

Per design, our hypothesis is that the EA-LSTM will learn to group similar basins together into the high-dimensional space, so that hydrologically-similar basins use similar parts of the LSTM cell states. This is dependent, of course, on the information content of the catchment attributes used as inputs, but the model should at least not degrade the quality of this information, and should learn hydrologic similarity in a way that is useful for rainfall-runoff prediction. We tested this hypothesis by analyzing the learned catchment embedding from a hydrological perspective. We analyzed geographical similarity by using k-means clustering on the $\mathbb{R}^{256}$ feature space of the input gate embedding to delineate basin groupings, and then plotted the clustering results geographically. The number of clusters was determined using a mean silhouette score.

In addition to visually analyzing the k-means clustering results by plotting them spatially (to ensure that the input embedding preserved expected geographical similarity), we measured the ability of these cluster groupings to explain variance in certain hydrological signatures in the CAMELS basins. For this, we used thirteen of the hydrologic signatures that were used by Addor et al. (2018): (i) mean annual discharge (q-mean), (ii) runoff ratio, (iii) slope of the flow duration curve (slope-fdc), (iv) baseflow index, (v) streamflow-precipitation elasticity (stream-elas), (vi) 5th percentile flow (q5), (vii) 95th percentile flow (q95), (viii) frequency of high flow days (high-q-freq), (ix) mean duration of high flow events (high-q-dur), (x) frequency of low flow days (low-q-freq), (xi) mean duration of low flow events (low-q-dur), (xii) zero flow frequency (zero-q-freq), and (xiii) average day of year when half of cumulative annual flow occurs (mean-hfd).

Finally, we reduced the dimension of the input gate embedding layer (from $\mathbb{R}^{256}$ to $\mathbb{R}^{2}$) so as to be able to visualize dominant features in the input embedding. To do this we used UMAP (McInnes et al., 2018), which is a nonparametric dimensionality reduction technique.

## 3 Results

This section is organized as follows:

– The first subsection (Sect. 3.1) presents a comparison between the three different LSTM-type model configurations discussed in Sect. 2.6.1. The emphasis in this comparison is to examine the effect of adding catchment attributes as additional inputs to the LSTM using the standard vs. adapted (EA-LSTM) architectures.

– The second subsection (Sect. 3.2) presents results from our benchmarking analysis – that is, the direct comparison between the performances of our EA-LSTM model with the full set of benchmark models outlined in Sect. 2.5.

– The third subsection (Sect. 3.3) present results of the sensitivity analysis outlined in Sect. 2.6.2.

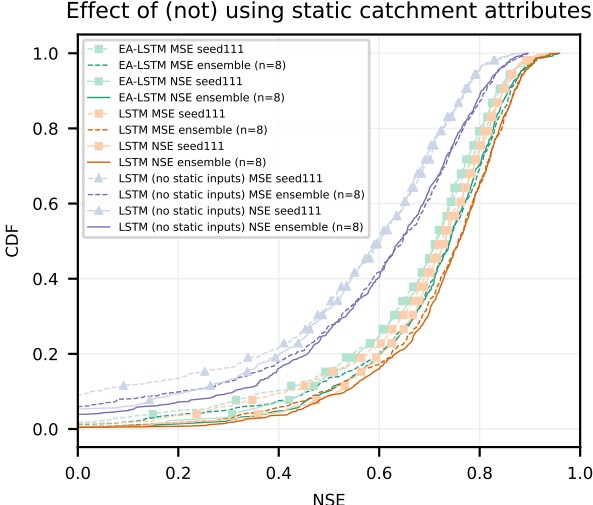

**Figure 3.** Cumulative density functions of the NSE for all LSTM-type model configurations described in Sect, 2.6.1. For each model type the ensemble mean and one of the $n = 8$ repetitions are shown. LSTM configurations are shown in orange (with catchment attributes) and purple (without catchment attributes), and the EA-LSTM configurations (always with catchment attributes) are shown in green.

- The final subsection (Sect. 3.4) presents an analysis of the EA-LSTM embedding layer to demonstrate that the model learned how to differentiate between different rainfall-runoff behaviors across different catchments.

### 3.1 Comparison between LSTM Modeling Approaches

The key results from a comparison between the LSTM approaches are in Fig. 3, which shows the cumulative density functions (CDF) of the basin-specific NSE values for all six LSTM models (three model configurations, and two loss functions) over the 531 basins.

Table 2 contains average key overall performance statistics. Statistical significance was evaluated using the paired Wilcoxon test (Wilcoxon, 1945), and the effect size was evaluated using Cohen's $d$ (Cohen, 2013). The comparison contains four key

results:

(i) Using catchment attributes as static input features improves overall model performance as compared with not providing the model with catchment attributes. This is expected, but worth confirming.

(ii) Training against the basin-average NSE* loss function improves overall model performance as compared with training against an MSE loss function, especially in the low NSE spectra.

(iii) There is statistically significant difference between the performance of the standard LSTM with static input features and the EA-LSTM however, with a small effect size.


(iv) Some of the error in the LSTM-type models is due to randomness in the training procedure and can be mitigated by running model ensembles.

Related to result (i), there was a significant difference between LSTMs with standard architecture trained with vs. without
static features (square vs. triangle markers in Fig. 3). The mean (over basins) NSE improved in comparison with the LSTM that did not take catchment characteristics as inputs by 0.44 (range (0.38, 0.56)) when optimized using the MSE and 0.30 (range(0.22, 0.43)) when optimized using the basin-average NSE*. To assess statistical significance for single models, the mean basin performance (e.g. mean NSE per basin and across all seeds) between two different model settings was compared between different model configurations. To assess statistical significance for ensemble means, the mean basin performance of
the ensemble mean was compared between different model configurations. For models trained using the standard MSE loss function, the p-value for the single model was $p = 1.2 * 10^{-75}$ and the p-value between the ensemble means was $p = 4 * 10^{-68}$. When optimized using the basin-average NSE*, the p-value for the single model was $p = 8.8 * 10^{-81}$) and the p-value between the ensemble means was $p = 3.3 * 10^{-75}$.

It is worth emphasizing that the improvement in overall model performance due to including catchment attributes implies
that these attributes contain information that helps to distinguish different catchment-specific rainfall-runoff behaviors. This is especially interesting since these attributes are derived from remote sensing and other everywhere-available data products, as described by Addor et al. (2017b). Our benchmarking analysis presented in the next subsection (Sect. 3.2), shows that this information content is sufficient to perform high quality regional modeling (i.e., competitive with lumped models calibrated separately for each basin).

Related to result (ii), using the basin-average NSE* loss function instead of a standard MSE loss function improved performance for single models (different individual seeds) as well as for the ensemble means across all model configurations (see Tab. 2). The differences are most pronounced for the EA-LSTM and for the LSTM without static features. For the EA-LSTM, the mean NSE for the single model raised from 0.63 when optimized with MSE to 0.67 when optimized with the basin average NSE*. For the LSTM trained without catchment characteristics the mean NSE went from 0.23 when optimized with MSE to
0.39 when optimized with NSE*. Further, the median NSE did not change significantly depending on loss function due to the fact that the improvements from using the NSE* are mostly to performance in basins at the lower-end of the NSE spectra (see also Figure 1 dashed vs. solid lines). This is as expected as catchments with relatively low average flows have a small influence on LSTM training with an MSE loss function, which results in poor performance in these basins. Using the NSE* loss function helps to mitigate this problem. It is important to note that this is not the only reason why certain catchments have low skill
scores, which can happen for a variety of reasons with any type of hydrological model (e.g., bad input data, unique catchment behaviors, etc.). This improvement at the low-performance end of the spectrum can also been seen by looking at the number of 'catastrophic failures', i.e., basins with an NSE value of less than zero. Across all models we see a reduction in this number when optimized with the basin average NSE*, compared to optimizing with MSE.

Related to result (iii), Fig. 3 shows a small difference in the empirical CDFs between the standard LSTM with static input
features and the EA-LSTM under both functions (compare green vs orange lines) . The difference is significant (p-value for single model $p = 1 * 10^{-28}$, p-value for the ensemble mean $p = 2.1 * 10^{-26}$, paired Wilcoxon test), however the effect size is





**Table 2.** Evaluation results of the single models and ensemble means.

| Model | NSE | | No. of basins |
| --- | --- | --- | --- |
| | mean | median | with NSE $\leq 0$ |
| **LSTM w/o static inputs** | | | |
| using MSE: | | | |
| Single model: | 0.24 ($\pm$ 0.049) | 0.60 ($\pm$ 0.005) | 44 ($\pm$ 4) |
| Ensemble mean (n=8): | 0.36 | 0.65 | 31 |
| using NSE*: | | | |
| Single model: | 0.39 ($\pm$ 0.059) | 0.59 ($\pm$ 0.008) | 28 ($\pm$ 3) |
| Ensemble mean (n=8): | 0.49 | 0.64 | 20 |
| **LSTM with static inputs** | | | |
| using MSE: | | | |
| Single model: | 0.66 ($\pm$ 0.012) | 0.73 ($\pm$ 0.003) | 6 ($\pm$ 2) |
| Ensemble mean (n=8): | 0.71 | 0.76 | 3 |
| using NSE*: | | | |
| Single model: | 0.69 ($\pm$ 0.013) | 0.73 ($\pm$ 0.002) | 2 ($\pm$ 1) |
| Ensemble mean (n=8): | 0.72 | 0.76 | 2 |
| **EA-LSTM** | | | |
| using MSE: | | | |
| Single model: | 0.63 ($\pm$ 0.018) | 0.71 ($\pm$ 0.005) | 9 ($\pm$ 1) |
| Ensemble mean (n=8): | 0.68 | 0.74 | 6 |
| using NSE*: | | | |
| Single model: | 0.67 ($\pm$ 0.006) | 0.71 ($\pm$ 0.005) | 3 ($\pm$ 1) |
| Ensemble mean (n=8): | 0.70 | 0.74 | 2 |

small $d = 0.055$. This is important because the embedding layer in the EA-LSTM adds a layer of interpretability to the LSTM, which we argue is desirable for scientific modeling in general, and is useful in our case for understanding catchment similarity. This is only useful, however, if the EA-LSTM does not sacrifice performance compared to the less interpretable traditional LSTM. There is some small performance sacrifice in this case, likely due to an increase in the number of tunable parameters in the network, but the benefit of this small reduction in performance is explainability.

Related to results (iv), in all cases there were several basins with very low NSE values (this is also true for the benchmark models, which we will discuss in Sect. 3.2). Using catchment characteristics as static input features with the EA-LSTM architecture reduced the number of such basins from 44 (31) to 9 (6) for the average single model (ensemble mean) when optimized with the MSE, and from 28 (20) to 3 (2) for the average single model (ensemble mean) if optimized using the basin-average



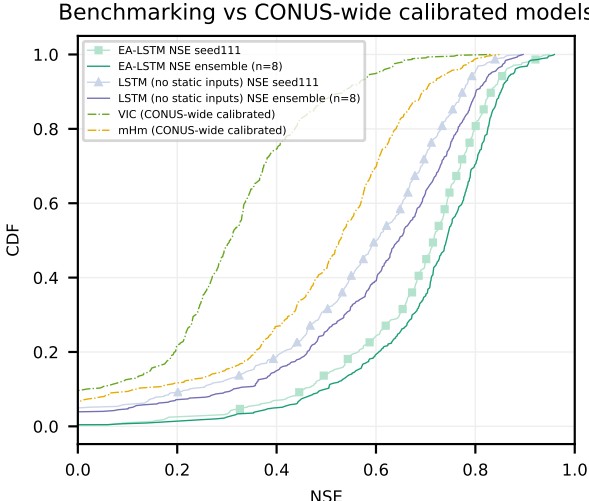

**Figure 4.** Cumulative density functions of the NSE of two regionally-calibrated benchmark models (VIC and mHM), compared to the EA-LSTM and the LSTM trained without static input features.

NSE*. This result is worth emphasizing: each LSTM or EA-LSTM trained over all basins results in a certain number of basins that perform poorly (NSE $\leq$ 0), but the basins where this happens are not always the same. The model outputs, and therefore the number of catastrophic failures, differ depending on the randomness in the weight initialization and optimization procedure and thus, running an ensemble of LSTMs substantively reduces this effect. This is good news for deep learning - it means that
at least a portion of uncertainty can be mitigated using model ensembles. We leave as an open question for future research how many ensemble members, as well as how these are initialized, should be used to minimize uncertainty for a given data set.

## 3.2    Model Benchmarking: Comparison with Traditional Calibrated Hydrology Models

The results in this section are calculated from 447 basins that were modeled by all benchmark models, as well as our LSTMs.

   First we compared the EA-LSTM against the two hydrological models that were regionally calibrated (VIC and mHM) .
Specifically, what was calibrated for each model was a single set of transfer functions that map from static catchment characteristics to model parameters. The procedure for parameterizing these models for regional simulations is described in detail by the original authors: Mizukami et al. (2017) for VIC and Rakovec et al. (2019) for mHM. Figure 4 shows that the EA-LSTM outperformed both regionally-calibrated benchmark models by a large margin. Even the LSTM trained without static catchment attributes (only trained on meteorological forcing data) outperformed both regionally calibrated models consistently as a
single model, and even more so as an ensemble.

   The mean and median NSE scores across the basins of the individual EA-LSTM models ($N_{ensemble} = 8$) were $0.67 \pm 0.006$ (0.71) and $0.71 \pm 0.004$ (0.74) respectively. In contrast, VIC had a mean NSE of 0.17 and a median NSE of 0.31 and the mHM had a mean NSE of 0.44 and median NSE of 0.53. Overall, VIC scored higher than the EA-LSTM ensemble in 2 out



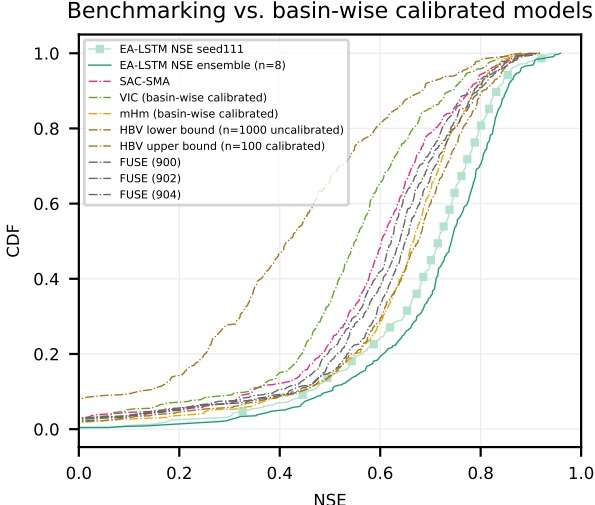

**Figure 5.** Cumulative density function of the NSE for all basin-wise calibrated benchmark models compared to the EA-LSTM.

of 447 basins (0.4%) and mHM scored higher than the EA-LSTM ensemble in 16 basins (3.58%). Investigating the number of

catastrophic failures (the number of basins where NSE $\leq$ 0), the average single EA-LSTM failed in approximately 2 basins out of 447 basins (0.4 $\pm$ 0.2%) and the ensemble mean of the EA-LSTM failed in only a single basin (i.e., 0.2%). In comparison mHM failed in 29 basins (6.49%) and VIC failed in 41 basins (9.17%).

Second, we compared our multi-basin calibrated EA-LSTMs to individual-basin calibrated hydrological models. This is a more rigorous benchmark than the regionally calibrated models, since hydrological models usually perform better when trained

for specific basins. Figure 5 compares CDFs of the basin-specific NSE values for all benchmark models over the 447 basins. Table 3 contains the performance statistics for these benchmark models as well as for the re-calculated EA-LSTM.

The main benchmarking result is that the EA-LSTM significantly outperforms all benchmark models in the overall NSE. The two best performing hydrological models were the ensemble ($n = 100$) of basin-calibrated HBV models and a single basin-calibrated mHM model. The EA-LSTM out-performed both of these models at any reasonable alpha level. The p-value

for the single model, compared to the HBV upper bound was $p = 1.9 * 10^{-4}$ and for the ensemble mean $p = 6.2 * 10^{-11}$ with a medium effect size (Cohen's $d$ for single model $d = 0.22$ and for the ensemble mean $d = 0.40$). The p-value for the single model, compared to the basin-wise calibrated mHM was $p = 4.3 * 10^{-6}$ and for the ensemble mean $p = 1.0 * 10^{-13}$ with a medium effect size (Cohen's $d$ for single model $d = 0.26$ and for the ensemble mean $d = 0.45$).

Regarding all other metrics except the Kling-Gupta decomposition of the NSE, there was no statistically significant differ-

ence between the EA-LSTM and the two best performing hydrological models. The $\beta$-decomposition of the NSE measures a scaled difference in simulated vs. observed mean streamflow values, and in this case the HBV benchmark performed better that the EA-LSTM, with an average scaled absolute bias (normalized by the root-variance of observations) of -0.01, where as the EA-LSTM had an average scaled bias of -0.03 for the individual model as well as for the ensemble ($p = 3.5 * 10^{-4}$).





**Table 3.** Comparison of the EA-LSTM average single model and ensemble mean to the full set of benchmark models. VIC (basin) and mHM (basin) denote the basin-wise calibrated models, while VIC (CONUS) and mHM (CONUS) denote the CONUS-wide calibrated models. HBV (lower) denotes the ensemble mean of $n = 1000$ uncalibrated HBVs, while HBV (upper) denotes the ensemble mean of $n = 100$ calibrated HBVs (for details see Seibert et al. (2018)). For the FUSE model, the numbers behind the name denote different FUSE model structures. All statistics were calculated from the validation period of all 447 commonly modeled basins.

| Model | NSE | | No. of basins | $\alpha$-NSE | $\beta$-NSE | FHV | FMS | FLV |
|---|---|---|---|---|---|---|---|---|
| | mean | median | with NSE $\leq 0$ | median | median | median | median | median |
| EA-LSTM Single | 0.674 | 0.714 | 2 | 0.82 | -0.03 | -16.9 | -10.0 | 2.0 |
| | ($\pm$ 0.006) | ($\pm$ 0.004) | ($\pm$ 1) | ($\pm$ 0.013) | ($\pm$ 0.009) | ($\pm$ 1.1) | ($\pm$ 1.7) | ($\pm$ 7.6) |
| EA-LSTM Ensemble | 0.705 | 0.742 | 1 | 0.81 | -0.03 | -18.1 | -11.3 | 31.9 |
| SAC-SMA | 0.564 | 0.603 | 13 | 0.78 | -0.07 | -20.4 | -14.3 | 37.3 |
| VIC (basin) | 0.518 | 0.551 | 10 | 0.72 | -0.02 | -28.1 | -6.6 | -70.0 |
| VIC (CONUS) | 0.167 | 0.307 | 41 | 0.46 | -0.07 | -56.5 | -28.0 | 17.4 |
| mHM (basin) | 0.627 | 0.666 | 7 | 0.81 | -0.04 | -18.6 | -7.2 | 11.4 |
| mHM (CONUS) | 0.442 | 0.527 | 29 | 0.59 | -0.04 | -40.2 | -30.4 | 36.4 |
| HBV (lower) | 0.237 | 0.416 | 35 | 0.58 | -0.02 | -41.9 | -15.9 | 23.9 |
| HBV (upper) | 0.631 | 0.676 | 9 | 0.79 | -0.01 | -18.5 | -24.9 | 18.3 |
| FUSE (900) | 0.587 | 0.639 | 12 | 0.80 | -0.03 | -18.9 | -5.1 | -11.4 |
| FUSE (902) | 0.611 | 0.650 | 10 | 0.80 | -0.05 | -19.4 | 9.6 | -33.2 |
| FUSE (904) | 0.582 | 0.622 | 9 | 0.78 | -0.07 | -21.4 | 15.5 | -66.7 |

## 3.3 Robustness and Feature Ranking

In Sect. 3.1, we found that adding static features provided a large boost in performance. We would like to check that the model is not simply 'remembering' each basin instead of learning a general relation between static features and catchment-specific hydrologic behavior. To this end, we examined model robustness with respect to noisy perturbations of the catchment attributes. Figure 6 shows the results of this experiment by comparing the model performance when forced (not trained) with perturbed static features in each catchment against model performance using the same static feature values that were used for training.

As expected, the model performance degrades with increasing noise in the static inputs. However, the degradation does not happen abruptly but smoothly with increasing levels of noise. To reiterate from Sect 2.6.2, the perturbation noise is always relative to the overall standard deviation of the static features across all catchments, which is always $\sigma = 1$ (i.e., all static input features were normalized prior to training). When noise with small standard deviation was added (e.g. $\sigma = 0.1$ and $\sigma = 0.2$) the mean and median NSEs were relatively stable. The median NSE decreased from 0.71 without noise to 0.48 with an added

noise equal to the total variance of the input features ($\sigma = 1$). This is roughly similar to the performance of the LSTM without static input features (Tab. 2). In contrast, the lower percentiles of the NSE distributions were more strongly affected by input



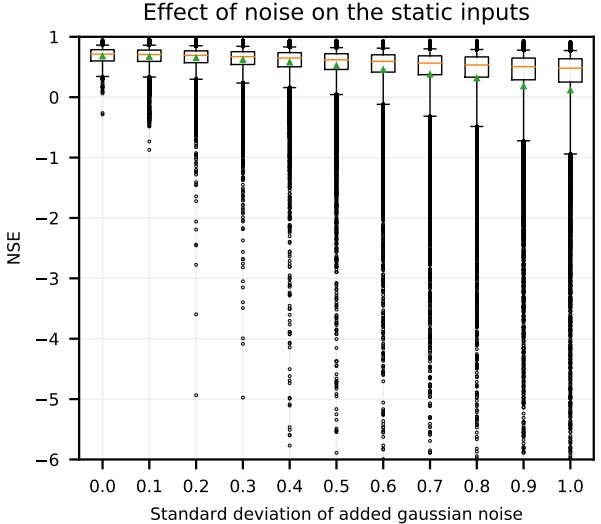

**Figure 6.** Boxplot showing degradation of model performance with increasing noise level added to the catchment attributes. Orange lines denote the median across catchments, green markers represent means across catchments, box denote the 25 and 75-percentiles, whiskers denote the 5 and 95-percentiles, and circles are catchments that fall outside the 5-95-percentile range.

noise. For example, the 1st (5th) percentile of the NSE values decreased from an NSE of 0.13 (0.34) to -5.87 (-0.94) when going from zero noise (the catchment attributes data from CAMELS) to additive noise with variance equal to the total variance of the inputs (i.e., $\sigma = 1$). This reinforces that static features are especially helpful for increasing performance in basins at the lower
end of the NSE spectrum - that is, differentiating hydrological behaviors that are under-represented in the training data set.

   Figure 7 plots a spatial map where each basin is labeled corresponding to the most sensitive catchment attribute derived from the explicit Morris method for neural networks (Sect. 2.6.2). In the Appilachain Mountains, sensitivity in most catchments is dominated by topological features (e.g., mean catchment elevation and catchment area), and in the Eastern US more generally, sensitivity is dominated by climate indices (e.g., mean precipitation, high precipitation duration). Meteorological patterns like
aridity and mean precipitation become more important as we move away from the Appalachians and towards the Great Plains, likely because elevation and slope begin to play less of a role. The aridity index dominates sensitivity in the Central Great Plains. In the Rocky Mountains most basins are sensitive climate indices (mean precipitation and high precipitation duration), with some sensitivity to vegetation in the Four-Corners region (northern New Mexico). In the West Coast there is a wider variety of dominant sensitivities reflecting a diversity of catchments.
Table 4 provides an overall ranking of dominant sensitivities. These were derived by normalizing the sensitivity measures per basin to the range (0,1) and then calculating the overall mean across all features. As might be inferred from Fig. 7 the most sensitive catchment attributes are topological features (mean elevation and catchment area) and climate indices (mean precipitation, aridity, duration of high precipitation events and the fraction of precipitation falling as snow). Certain groups of



Highest ranked feature per basin

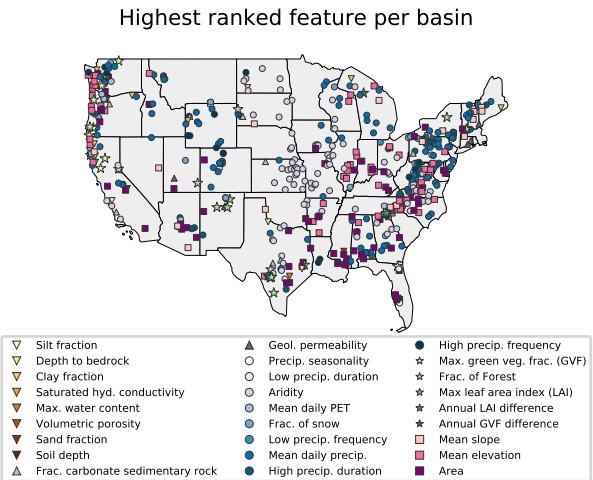

**Figure 7.** Spatial map of all basins in the data set. Markers denote the individual catchment characteristic with the highest sensitivity value for each particular basin.

catchment attributes did not typically provide much additional information. These include vegetation indices like maximum

leaf area index or maximum green vegetation fraction, as well as the annual vegetation differences. Most soil features were at the lower end of the feature ranking. This sensitivity ranking is interesting in that most of the top-ranked features are relatively easy to measure or estimate globally from readily-available gridded data products. Soil maps are one of the hardest features to obtain accurately at a regional scale because they require extensive in situ mapping and interpolation. It is worth noting that our rankings qualitatively agree with much of the analysis by Addor et al. (2018).

**3.4    Analysis of Catchment Similarity from the Embedding Layer**

Kratzert et al. (2018a, 2019) showed that these LSTM networks are able to learn to model snow, and store this information in specific memory cells, without ever directly training on any type of snow-related observation data other than total precipitation and temperature. Multiple types of catchments will use snow-related states in mixture with other states that represent other processes or combinations of processes. The memory cells allow an interpretation along the time axis for each specific basin,

and are part of both the standard LSTM and the EA-LSTM. A more detailed analysis of the specific functionality of individual cell states is out-of-scope for this manuscript and will be part of future work. Here, we focus on analysis of the embedding layer, which is a unique feature of the EA-LSTM.

From each of the trained EA-LSTM models, we calculated the input gate vector (Eq. 7) for each basin. The raw EA-LSTM embedding from one of the models trained over all catchments is shown in Fig. 8. Yellow colors indicate that a particular

one of the 256 cell states is activated and contributes to the simulation of a particular catchment. Blue colors indicate that a





**Table 4.** Feature ranking derived from the explicit Morris method.

| Rank | Catchment characteristic | Sensitivity |
|---|---|---|
| 1. | Mean precipitation | 0.68 |
| 2. | Aridity | 0.56 |
| 3. | Area | 0.50 |
| 4. | Mean elevation | 0.46 |
| 5. | High precip. duration | 0.41 |
| 6. | Fraction of snow | 0.41 |
| 7. | High precip. frequency | 0.38 |
| 8. | Mean slope | 0.37 |
| 9. | Geological permeability | 0.35 |
| 10. | Frac. carbonate sedimentary rock | 0.34 |
| 11. | Clay fraction | 0.33 |
| 12. | Mean PET | 0.31 |
| 13. | Low precip. frequency | 0.30 |
| 14. | Soil depth to bedrock | 0.27 |
| 15. | Precip. seasonality | 0.27 |
| 16. | Frac. of Forest | 0.27 |
| 17. | Sand fraction | 0.26 |
| 18. | Saturated hyd. conductivity | 0.24 |
| 19. | Low precip. duration | 0.22 |
| 20. | Max. green veg. frac. (GVF) | 0.21 |
| 21. | Annual GVF diff. | 0.21 |
| 22. | Annual leaf area index (LAI) diff. | 0.21 |
| 23. | Volumetric porosity | 0.19 |
| 24. | Soil depth | 0.19 |
| 25. | Max. LAI | 0.19 |
| 26. | Silt fraction | 0.18 |
| 27. | Max. water content | 0.16 |

particular cell state is not used for a particular catchment. These (real-valued) activations are a function of the 27 catchment characteristics input into the static feature layer of the EA-LSTM.

The embedding layer is necessarily high-dimensional – in this case $\mathbb{R}^{256}$ – due to the fact that the LSTM layer of the model requires sufficient cell states to simulate a wide variety of catchments. Ideally, hydrologically-similar catchments should utilize





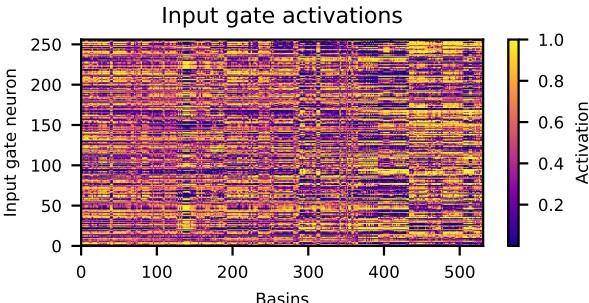

**Figure 8.** Input gate activations (y-axis) for all 531 basins (x-axis). The basins are ordered from left to right according to the ascending 8-digit USGS gauge ID. Yellow colors denote open input gate cells, blue colors denote closed input gate cells for a particular basin.

overlapping parts of the LSTM network - this would mean that the network is both learning and using catchment similarity to train a regionalizable simulation model.

To assess whether this happened, we first performed a clustering analysis on the $\mathbb{R}^{256}$ embedding space using k-means with an Euclidean distance criterion. We compared this with a k-means clustering analysis using directly the 27 catchment characteristics, to see if there was a difference in clusters before vs. after the transformation into the embedding layer - remember that

this transform was informed by rainfall-runoff training data. To choose an appropriate cluster size, we looked at the mean (and minimum) silhouette value. Silhouette values measure within-cluster similarity and range between [-1,1], with positive values indicating a high degree of separation between clusters, and negative values indicating a low degree of separation between clusters. The mean and minimum silhouette values for different cluster sizes are shown in Fig. 9. In all cases with cluster sizes less than 15, we see that clustering by the values of the embedding layer provides more distinct catchment clusters than

when clustering by the raw catchment attributes. This indicates that the EA-LSTM is able to use catchment attribute data to effectively cluster basins into distinct groups.

The highest mean silhouette value from clustering with the raw catchment attributes was $k = 6$ and the highest mean silhouette value from clustering with the embedding layer was $k = 5$. Ideally, these clusters would be related to hydrologic behavior. To test this, Fig. 10 shows the fractional reduction in variance of 13 hydrologic signatures due to clustering by both raw

catchment attributes vs. by the EA-LSTM embedding layer. Ideally, the within-cluster variance of any particular hydrological signature should be as small as possible, so that the fractional reduction in variance is as large (close to one) as possible. In both the $k = 5$ and $k = 6$ cluster examples, clustering by the EA-LSTM embedding layer reduced variance in the hydrological signatures by more or approximately the same amount as by clustering on the raw catchment attributes. The exception to this was the hfd-mean date, which represents an annual timing process (i.e., the day of year when the catchment releases

half of its annual flow). This indicates that the EA-LSTM embedding layer is largely preserving the information content about hydrological behaviors, while overall increasing distinctions between groups of similar catchments. The EA-LSTM was able to learn about hydrologic similarity between catchments by directly training on both catchment attributes and rainfall-runoff



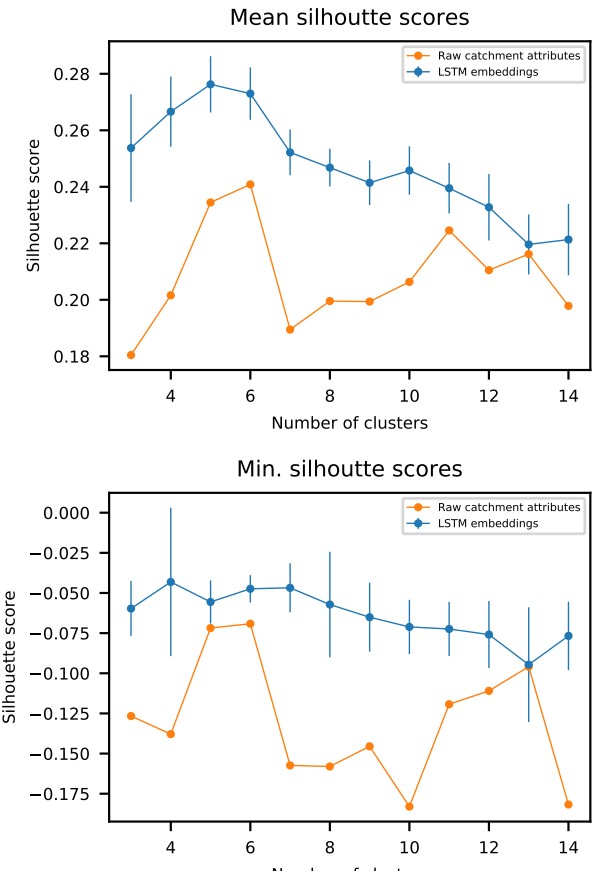

**Figure 9.** Mean and minimum silhouette scores over varying cluster sizes. For the LSTM embeddings, the line denotes the mean of the $n = 8$ repetitions and the vertical lines the standard deviation over 10 random restarts of the k-means clustering algorithm.

time series data. Remember that the EA-LSTMs were trained on the time series of streamflow data that these signatures were calculated from, but were not trained directly on these hydrologic signatures.

Clustering maps for $k = 5$ and $k = 6$ are shown in Fig. 11. Although latitude and longitude were not part of the catchment attributes vector that was used as input into the embedding layer, both the raw catchment attributes and the embedding layer clearly delineated catchments that correspond to different geographical regions within the CONUS.

To visualize the high-dimensional embedding learned by the EA-LSTM, we used UMAP (McInnes et al., 2018) to project the full $\mathbb{R}^{256}$ embedding onto $\mathbb{R}^2$. Figure 12 shows results of the UMAP transformation for one of the eight EA-LSTMs. In 475    each subplot in Fig. 12, each point corresponds to one basin. The absolute values of the transformed embedding are not of particular interest, but we are interested in the relative arrangement of the basins in this 2-dimensional space. Because this is a reduced-dimension transformation, the fact that there are three clear clusters of basins does not necessarily indicate that



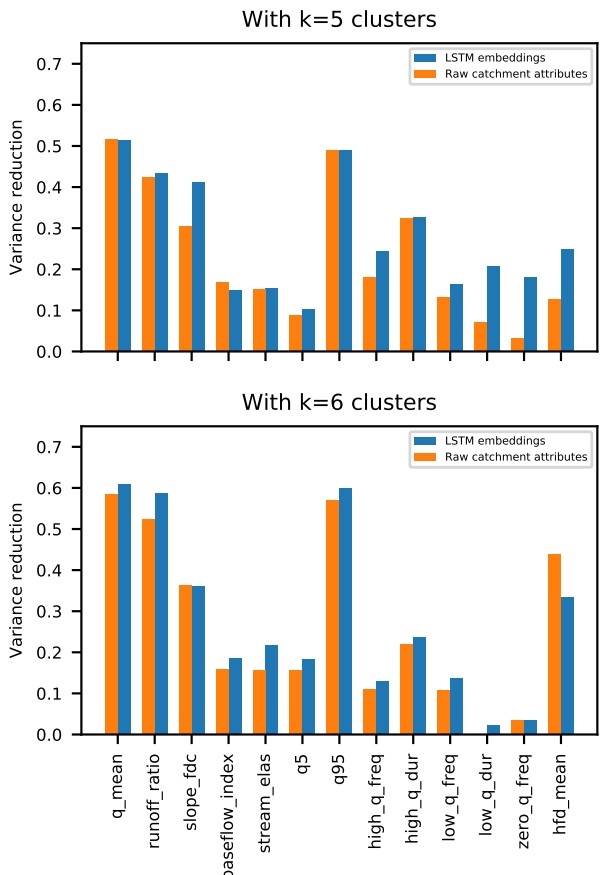

**Figure 10.** Fractional reduction in variance about different hydrological signatures due to k-means clustering on catchment attributes vs. the EA-LSTM embedding layer.

these are the only distinct basin clusters in our 256-dimensional embedding layer (as we saw above). Figure 12 shows that there is strong interaction between the different catchment characteristics in this embedding layer. For example, high-elevation
dry catchments with low forest cover are in the same cluster as low-elevation wet catchments with high forest cover. These two groups of catchments share parts of their network functionality in the LSTM, whereas highly seasonal catchments activate a different part of the network. Additionally, there are two groups of basins with a high forest fractions, however if we also consider the mean annual green vegetation difference, both of these clusters are quite distinct. The cluster in the upper left of each subplot in Fig. 12 contains forest type basins with a high annual variation in the green vegetation fraction (possibly
deciduous forests) and the cluster at the right has almost no annual variation (possibly coniferous forests). One feature that does not appear to affect catchment groupings (i.e., apparently acts independent of other catchment characteristics) is basin size – large and small basins are distributed throughout the three UMAP clusters. To summarize, this analysis demonstrates that the EA-LSTM is able to learn complex interactions between catchment attributes that allows for grouping different basins (i.e.,





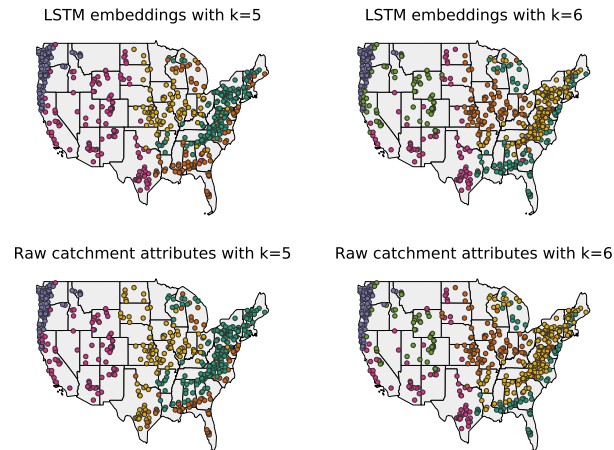

**Figure 11.** Clustering maps for the LSTM embeddings (top row) and the raw catchment attributes (bottom row) using $k = 5$ clusters (left column, optimal choice for LSTM embeddings) and $k = 6$ clusters (right column, optimal choice for the raw catchment attributes)

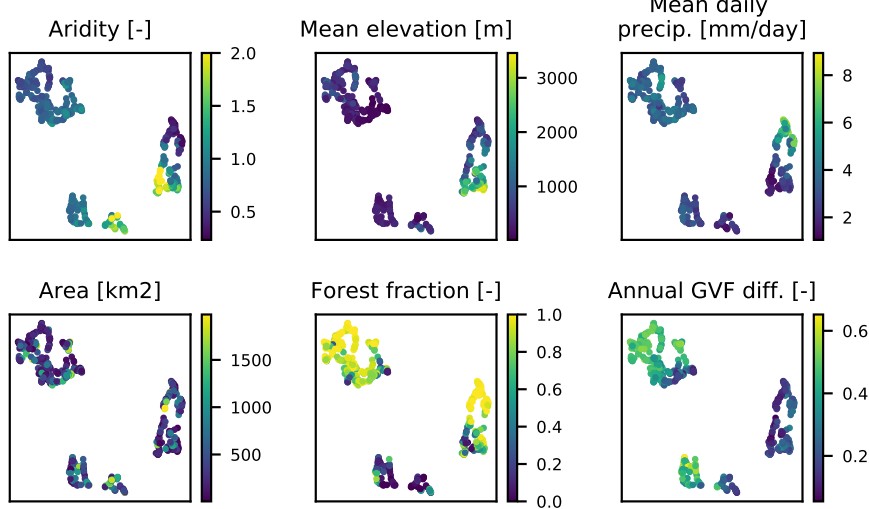

**Figure 12.** UMAP transformation of the $\mathbb{R}^{256}$ EA-LSTM catchment embedding onto $\mathbb{R}^2$. Each dot in each subplot corresponds to one basin. The colors denote specific catchment attributes (notated in subplot titles) for each particular basin.

choosing which cell states in the LSTM any particular basin or group of basins will use) in ways that account for interaction

between different catchment attributes.





## 4 Discussion and Conclusion

The EA-LSTM is an example of what Razavi and Coulibaly (2013) called a *model-independent* method for regional modeling. We cited Besaw et al. (2010) as an earlier example this type of approach, since they also used classical feed-forward neural networks. In our case, the EA-LSTM achieved state-of-the-art results, outperforming multiple locally- and regionally-calibrated benchmark models. These benchmarking results are arguably the critical result of this paper.


The innovation in this study – besides benchmarking the LSTM family of rainfall-runoff models – was to add a static embedding layer in the form of our EA-LSTM. This model offered similar performance as compared with a conventional LSTM (Sect. 3.1) but offers a level of interpretabiltiy about how the model learns to differentiate aspects of complex catchment-specific behaviors (Sect. 3.3 and Sect. 3.4). In a certain sense, this is similar to the aforementioned MPR approach, which links its model parameters to the given spatial characteristics (in a non-linear way, by using transfer functions), but has a fixed model structure to work with. In comparison, our EA-LSTM links catchment characteristics to the dynamics of specific sites and learns the overall model from the combined data of all catchments.


Neural networks generally require a lot of training data (our unpublished results indicate that it is often difficult to reliably train an LSTM at a single catchment, even with multi-decade data records), and adding the ability for the LSTM architecture to transfer information from similar catchments is critical for this to be a viable approach for regional modeling. This is in contrast with traditional hydrological modeling and model calibration, which typically has the best results when models are calibrated independently for each basin. This property of classical models is somewhat problematic, since it has been observed that the spatial patterns of model parameter obtained by ad-hoc extrapolations based on calibrated parameters from reference catchments can lead to unrealistic parameter fields and spatial discontinuities of the hydrological states (Mizukami et al., 2017). As shown in Sect. 3.4 this does not occur with our proposed approach. Thus, by leveraging the ability of deep learning to simultaneously learn time series relationships and also spatial relationships in the same predictive framework, we sidestep many problems that are currently associated with the estimation and transfer of hydrologic model parameters.



Moving forward, however, it is worth mentioning that treating catchment attributes as static is a strong assumption (especially over long time periods), which is not necessarily reflected in the real world. In reality, catchment attributes may continually change at various timescales (e.g., vegetation, topography, pedology, climate). In future studies it will be important to develop strategies to derive analogues to our embedding layer that allow for dynamic or evolving catchment attributes or features - perhaps that act on raw remote sensing data inputs rather than aggregated indexes derived from time series of remote sensing products. In principle, our embedding layer could learn directly from raw brightness temperatures, since there is no requirement that the inputs be hydrologically relevant - only that these inputs are related to hydrological behavior. A dynamic input gate is, at least in principle, possible without significant modification to the proposed EA-LSTM approach. For example, by using a separate sequence-to-sequence LSTM that encodes time-dependent catchment observables (e.g., from climate models or remote sensing) and feeds an embedding layer that is updated at each timestep. This would allow the model to 'learn' a dynamic embedding that turns off and on different parts of the rainfall-runoff portion of the LSTM over the course of a simulation.





A notable corollary of our main result is that the catchment attributes collected by Addor et al. (2017b) appear to contain sufficient information to distinguish between diverse rainfall-runoff behaviors, at least to a meaningful degree. It is arguable whether this was known previously, since regional modeling studies have largely struggled to fully extract this information (Mizukami et al., 2017) - i.e., existing regional models do not perform with accuracy similarly to models calibrated in a specific catchment. In contrast, our regional EA-LSTM actually performs better than models calibrated separately for individual catchments. This result goes somewhat against the commonly-held belief that runoff time series alone only bear enough information to restrict a handful of parameters (Naef, 1981; Jakeman and Hornberger, 1993; Perrin et al., 2001; Kirchner, 2006), and implies that structural improvements are still possible for most large-scale hydrology models, given the size of today's data sets.

*Code and data availability.* The CAMELS input data is freely available at the homepage of NCAR. The validation period of all benchmark models used in this study is available at DOI. The code to reproduce the results of this manuscript can be found at https://github.com/kratzert/ealstm_regional_modeling

*Author contributions.* FK had the idea for for the regional modeling approach. SH proposed the adapted LSTM architecture. FK, DK and GN designed all experiments. FK conducted all experiments and analysed the results, together with DK, GS and GN. GN supervised the manuscript from the hydrological perspective and GK and SH from the machine learning perspective. GN and SH share the responsibility for the last-authorship in the respective fields. All authors worked on the manuscript.

*Competing interests.* The authors declare that they have no conflict of interest.

*Acknowledgements.* The project relies heavily on open source software. All programming was done in Python version 3.7 (van Rossum, 1995) and associated libraries including: Numpy (Van Der Walt et al., 2011), Pandas (McKinney, 2010), PyTorch (Paszke et al., 2017) and Matplotlib (Hunter, 2007)

This work was supported by Bosch, ZF, and Google. We thank the NVIDIA Corporation for the GPU donations, LIT with grant LIT-2017-3-YOU-003 and FWF grant P 28660-N31.



## Appendix A: Full list of the used CAMELS Catchment Characteristics

**Table A1.** Table of catchment attributes used in this experiments. Description taken from the data set Addor et al. (2017a)

| | |
|---|---|
| p_mean | Mean daily precipitation. |
| pet_mean | Mean daily potential evapotranspiration. |
| aridity | Ratio of mean PET to mean precipitation. |
| p_seasonality | Seasonality and timing of precipitation. Estimated by representing annual precipitation and temperature as sin waves. Positive (negative) values indicate precipitation peaks during the summer (winter). Values of approx. 0 indicate uniform precipitation throughout the year. |
| frac_snow_daily | Fraction of precipitation falling on days with temperatures below $0^{\circ}C$. |
| high_prec_freq | Frequency of high precipitation days (>= 5 times mean daily precipitation). |
| high_prec_dur | Average duration of high precipitation events (number of consecutive days with >= 5 times mean daily precipitation). |
| low_prec_freq | Frequency of dry days (< 1 mm/day). |
| low_prec_dur | Average duration of dry periods (number of consecutive days with precipitation < 1 mm/day). |
| elev_mean | Catchment mean elevation. |
| slope_mean | Catchment mean slope. |
| area_gages2 | Catchment area. |
| forest_frac | Forest fraction. |
| lai_max | Maximum monthly mean of leaf area index. |
| lai_diff | Difference between the max. and min. mean of the leaf area index. |
| gvf_max | Maximum monthly mean of green vegetation fraction. |
| gvf_diff | Difference between the maximum and minimum monthly mean of the green vegetation fraction. |
| soil_depth_pelletier | Depth to bedrock (maximum 50m). |
| soil_depth_statsgo | Soil depth (maximum 1.5m). |
| soil_porosity | Volumetric porosity. |
| soil_conductivity | Saturated hydraulic conductivity. |
| max_water_content | Maximum water content of the soil. |
| sand_frac | Fraction of sand in the soil. |
| silt_frac | Fraction of silt in the soil. |
| clay_frac | Fraction of clay in the soil. |
| carb_rocks_frac | Fraction of the catchment area characterized as "Carbonate sedimentary rocks". |
| geol_permeability | Surface permeability (log10). |





## Appendix B: Hyperparameter tuning

The hyperparameters, i.e., the number of hidden/cell states, dropout rate, length of the input sequence and the number of stacked LSTM layers for our model were found by running grid search over a range of parameter values. Concretely we considered the following possible parameter values:

1. Hidden states: 64, 96, 128, 156, 196, 224, 256

2. Dropout rate: 0.0, 0.25, 0.4, 0.5

3. Length of input sequence: 90, 180, 270, 365

4. Number of stacked LSTM layer: 1, 2

We used k-fold cross validation ($k = 4$) to split the basins into an a training set and an independent test set. We trained one model for each split for each parameter combination on the combined calibration period of all basins in the specific training set and evaluated the model performance on the calibration data of the test basins. The final configuration was chosen by taking the parameter set that resulted in the highest median NSE over all possible parameter configurations. The parameters are:

1. Hidden states: 256

2. Dropout rate: 0.4

3. Length of input sequence length: 270

4. Number of stacked LSTM layer: 1





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
