# Peer review of "Towards Learning Universal, Regional, and Local Hydrological Behaviors via Machine-Learning Applied to Large-Sample Datasets"

_Hydrology and Earth System Sciences, 2019_

## Referee Comment (RC1) · Anonymous Referee #1 · 28 Aug 2019

This very interesting paper of Kratzert, et al. compares the quality of the predictions of various hydrological models with three variants of the Long Short-Term Memory (LSTM) deep learning network. One of these variants, the novel EA-LSTM, is trained using meteorological data and catchment similarities as an additional input and is analysed in detail highlighting the superiority of such a network. In general the paper is very well written and it is worth to be published after some minor changes. Some comments:

1. Maybe you could explain better the differences of the analysis of the single model and the ensemble mean approach. On page 13, lines 317-320 you write: "To assess statistical significance for single models, the mean basin performance (e.g. mean NSE

per basin and across all seeds) between two different model settings was compared between different model configurations." What's the difference between model settings and configuration? If I understood it correctly the difference in the verification of the single models and the ensemble mean is: Single model: From 8 ensemble model runs, you get 8 different predictions and you calculate the verification measures (e.g. NS values) for each of it and take the average (+/- Std? in Table 2); whereas in the Ensemble mean approach, for example this measure is calculated taking the mean of the 8 predictions? 2. For clarity reasons I would not include the single model outcome in Figure 3, because this a random outcome and would look different for each ensemble run.

3. Nice to have the significance reported, which is most often not shown. Although the precision of these p-values is extremely high and the differences are probably rather neglectable caused by noise.

4. Regarding the modified NSE. Wouldn't it be easier to normalize the streamflow data (e.g. using the BoxCox transformation)? So you don't have to event a new measure and adding a constant in order to achieve stable results.

5. Looking at the results, I would conclude that the EA-LSTM is very interesting for this analysis, but for practical applications the LSTM with the coupled meteo data and catchment attributes is even more efficient and is less complex. That's why I would like to see the results of this model also in Figure 4, 5 and Table 3.

6. Are the catchment attributes kept static for all days of the year? For example the monthly mean of leaf area index could be easily varied depending on the month of the year?

7. I would suggest to delete the UMAP analysis, since the method is not explained and the results are a bit confusing.

I'm looking forward to see the results of your planned work with a higher number of

ensemble members and the inclusion of the dynamic and the static input in the EA-LSTM input layer!

---

## Referee Comment (RC2) · Hoshin Gupta (Referee) · 10 Sep 2019

"Benchmarking a Catchment-Aware Long Short-Term Memory Network (LSTM) for Large-Scale Hydrological Modeling" by F Kratzert, D Klotz, G Shalev, G Klambauer, S Hochreiter and G Nearing Review by Hoshin Gupta

Brief Summary of the Paper: [1] This paper presents a novel approach to the problem of catchment-scale modeling.

[2] The classical "hydrological community" approach is to develop conceptual models (CM) of catchment-scale input-state-output behavior that have fixed/rigid structures

and parameterizations – that reflect our "scientific/physical" understanding of internal catchment structure (architecture) and functioning (processes and their interactions) – and to then apply these rigid pre-specified structures to different locations by altering the values of the (largely empirical) static parameters that are initially left unspecified (except typically to within "feasible ranges").

[3] Major challenges associated with this CM approach have been discussed in the literature, including the fact that the proposed rigid model structures are difficult to update/correct based on their inability to reproduce observed input-output dynamics sufficiently well (Bulygina and Gupta 2009, 2010, 2011; Gupta and nearing 2014; nearing and Gupta 2015), and that the free (optimizable) static parameters of such models have proven challenging to regionalize or relate to observable static data that is expected, based on hydrological understanding, to be (directly or indirectly) informative regarding differences in catchment functioning at different locations.

[4] In contrast, the authors use a machine-learning (ML) approach, based in Long Short-Term Memory Networks, that enables learning, from time-series input-output data, the system structural patterns associated with the observed dynamical system behaviors. So, while classical catchment CM's have "universal structural forms" that have been posed as hypotheses by scientists observing numerous examples of catchments across the world (or across a give region), the ML approach presented herein actually detects and learns the dynamics related attributes of such a universal catchment structure by being given access to time-series data from a great many such catchments.

[5] So, while classical catchment models are highly regularized (structurally constrained) using prior knowledge and the only remaining learning problem is to find values for the model parameters, the machine-learning approach presented herein must both learn the appropriate system structure and the appropriate location specific parameters necessary for the resulting model to provide good location specific performance. The lack of strong prior regularization means that such models cannot be

meaningfully trained on individual catchments and expected to give good performance, because the information necessary to unambiguously learn the "dynamical principles of catchment-scale hydrological behavior" are generally not going to be readily available in any single catchment data set.

[6] In a previous paper [Rainfall–runoff modelling using Long Short-Term Memory (LSTM) networks, HESS 2018], the authors demonstrated that the LSTM type of artificial neural network is suitable for catchment-scale hydrological modeling because of its ability to learn the long-term input-output dependencies that are essential for modelling the storage effects in catchments (e.g., snow accumulation and melt). They also demonstrated that such an approach can be used for regional scale modeling, where a single learned ML model can be used to simulate the discharge at a variety of catchments, and that the single ML models encoding of process behavior at the regional scale actually helps to improve model performance at each individual catchment through transfer learning (i.e., the multiple-catchment data helps to regularize the problem, so that the broader knowledge of catchment-scale behavior serves to improve the stability of local catchment-scale simulations/predictions).

[7] In this paper, the authors extend on that work to:

(1) Demonstrate that such an LSTM can be adapted to be able to capitalize on the availability of observable ancillary data in the form of catchment attributes to produce accurate streamflow estimates over a large number of basins.

(2) Show that the ML model can provide statistically significantly better performance (across a large number of catchments) than several existing CM type hydrology models that embed prior knowledge regarding catchment hydrological structure

(3) Demonstrate the way in which the ML model makes use of information in the ancillary data about catchment characteristics to differentiate between different rainfall-runoff behaviors, thereby enabling the superior performance obtained.

[8] To do so, the authors test two approaches, one in which the LSTM-based model is provided data regarding static catchment attributes as additional inputs at every time step (requiring no modification to the typical LSTM architecture) , and a second that is developed as a modification to LSTM architecture in which the data regarding static catchment attributes is provided separately in a manner that controls (through the input gate) which parts of the LSTM structure are activated for any individual catchment. The call the latter an Entity-Aware-LSTM (EA-LSTM) because it explicitly differentiates between similar types of dynamical behaviors that differ between individual entities (watersheds).

[9] The second (EA-LSTM) approach also differs from the first one in that it allows direct posterior inspection of the ML-based model structure to investigate what the model has actually learned from the static catchment attributes. The authors do this by investigating the nature of the mapping from catchment attribute space into the ML-model learned embedding space in which catchments with similar rainfall-runoff behavior are clustered together, thereby facilitating a (data-driven) catchment similarity analysis.

[10] In brief, the authors show that:

(1) Both the LSTM and EA-LSTM statistically outperformed the regionally-calibrated CM-type benchmark models by a large margin, as assessed using the NSE performance metric.

(2) The multi-basin calibrated EA-LSTM even (statistically) outperformed the individual-basin-calibrated hydrological models (a more rigorous benchmark), as assessed using the NSE performance metric.

(3) The use of catchment attributes as static input features significantly improves overall ML-based model performance as compared with when the model is not provided with information regarding catchment attributes. While an anticipated finding, the demonstration is both satisfying and convincing.

(4) The newly proposed EA-LSTM approach provides much better interpretability and potential contributions to hydrological understanding and insight regarding catchment similarity compared to the less interpretable traditional LSTM, without significant sacrifice of performance as assessed using the NSE performance metric.

(5) The large boost in ML-model performance obtained by providing information regarding static catchment features is not simply due to 'remembering' each basin instead of learning a general relation between static input features and catchment specific hydrologic behavior. Adding noise to the catchment attribute data causes only gradual deterioration in predictive performance. Further, striking improvements are seen for basins at the lower end of the performance spectrum which largely represent catchment types that are under-represented in the training data set.

(6) Regional differences in catchment behavior sensitivity to catchment attributes seems consistent with prior hydrological understanding (topography in the Appalachian Mountains, climate indices in the Eastern US, meteorological patterns as we move away towards the Great Plains, etc.). This observed sensitivity ranking is encouraging because most of the top-ranked features are relatively easy to measure or estimate globally from readily-available gridded data products.

(7) Certain groups of catchment attributes did not typically provide much additional information → these included vegetation indices like maximum leaf area index or maximum green vegetation fraction, as well as the annual vegetation differences. Further, most of the soil-related attributes were at the lower end of the feature ranking; this is interesting because soil characteristics are among the hardest features to characterize accurately at a regional scale.

(8) Clustering of "similar" catchments by the values of the EA-LSTM embedding layer provides more distinct results than when clustering by the raw catchment attributes, and seems to be well related to hydrologic behaviors, as assessed in terms of a set of 13 hydrologic signatures, indicating that the EA-LSTM embedding layer largely preserves the information content about hydrological behaviors, while overall increasing distinctions between groups of similar catchments. Further, the EA-LSTM seems able to learn complex interactions between catchment attributes that allows for grouping different basins in ways that account for interactions between different catchment attributes.

[11] In addition, the authors demonstrate that when training ML models to learn system structure regarding dynamical catchment behavior from large data sets (large numbers of catchments), it is important to account for the achievable differences in model performance at each catchment by adjusting the training performance metric. In this regard they propose a modified NSE loss function that seeks to account for the differences in means and variances of the observation data across basins, and that the performance (as assessed using MSE) is typically smaller for basins with low average discharge. By using the average of the NSE values at each basin that supplies training data as the ML-model training metric (referred to as the NSE*), the authors show that:

(9) Training against the basin-average NSE* loss function improves overall ML-model performance as compared with training against an MSE loss function, especially in the low NSE spectra. In particular there is a significant reduction in the number of basins that are classified as 'catastrophic failures' (i.e., basins with an NSE value of less than zero).

(10) Because the model outputs, and therefore the number of catastrophic failures, differ depending on the randomness in the weight initialization and optimization procedure, running an ensemble of LSTMs can substantively reduce this effect. My Assessment of the Paper:

[12] I believe that this paper represents a very significant contribution to the Earth System literature related to the development of Dynamical Environmental Systems Models (DESMs). I have alluded to some of the problems associated with the conventional CM approach in paragraphs [2-6] above. In this regard, there has been increasing commu-

nity interest in the use of both "large sample" data sets and the use of "model-structural-correction-via-data-assimilation" (learning from data) to extract better understanding about the structure and functioning of hydrological systems, such as catchments.

[13] This paper bridges the challenges of learning from large sample data sets and learning how catchments structures/behaviors can differ at local to regional scales in a very meaningful way. While not addressing the problem of prediction in un-gaged basins directly, the ability of the EA-LSTM to learn from and characterize differences in catchment functioning encoded in catchment attribute data is highly significant, and it would seem that a natural next step would be for the authors to demonstrate that potential by running experiments that seek to demonstrate that predictive ability learned from gaged locations can be transferred to un-gaged locations. I look forward to reading more about this in the future.

[14] As such, I have only a few suggestions to offer the authors. The first is that the current title "Benchmarking a Catchment-Aware Long Short-Term Memory Network (LSTM) for Large-Scale Hydrological Modeling" presents a rather technical front to what is arguably (in my opinion) a much more significant piece of work. I therefore offer up the possibility for the authors to consider that the introduction and discussion/conclusions sections be somewhat revamped/broadened to reflect the perspectives offered in my above summary of the paper. As indicated, I do think this paper is really more about the interesting challenges of learning and characterizing (via dynamical systems models) the "behavior and functioning" of hydrological systems at the catchment scale in such a manner that both universal (fundamentally hydrological) principles, and local-to-regional scale uniquenesses of such systems can be learned by accessing the patterns of information encoded in large sample data sets (Gupta et al 2014). In this regard the title could also then be generalized to reflect the nature of the conversation about "Learning Universal, Regional and Local Hydrological Behaviors via Machine-Learning applied to Large Sample data Sets". Or this more general discussion could be saved for a future publication .

[15] The second is that while the basin-average NSE* loss function does seem to serve the immediate needs of this study, I think that the ML-approach (and more generally hydrological learning from catchment data sets) can benefit from a more thoughtful approach to the problem of model performance metrics. In particular, the use of the observed output data "mean" as a benchmark for constructing the NSE itself, and the use of the output data variance to "normalize" across catchments to obtain somewhat comparable metric values to be averaged (or otherwise summarized in some statistical manner) seems, to me, problematic. In this regard, I think an Information Theoretic approach might ultimately prove to be more meaningful. I point out that the value of the metric, when used as the basis for assessing across different catchment locations, would be much enhanced if it somehow recognized the relative differences in complexity/difficulty associated with modeling the dynamical input-state-output behaviors at different locations (due to climatic, geological, and other factors). As discussed by Schaefli and Gupta (2007), the problem is at least partly one of appropriate benchmarking in order to make metric values meaningfully comparable. Some types of catchments (such as humid ones perhaps) are relatively easy to model to the level of obtaining high performance (e.g. NSE) values, while others (such as arid ones perhaps) are much more difficult to model … potentially requiring more complex model structures, more data, and perhaps better data quality. Since the challenge here is learning hydrological principles from the data, and some catchment systems are easier to characterize using simpler model structures, it would seem prudent to figure out how to account for this knowledge in the designs of our learning systems, which includes the metrics used as the filter through which information is being extracted.

[16] Finally, I think that the aforementioned issue may also relate to the fact that certain catchment attributes tend to be dominant indicators of differences in catchment behaviors, while others seem to show "lower importance" (sensitivity). It is been well known that "climate" (and one would reasonably expect also "topography") is the dominant indicator of catchment similarity, but this does not really help us to understand what structural differences in catchments drive differences in their behaviors. The finding

that soil and vegetation characteristics are low on the "importance" list is interesting, as it suggests that the existing catchment attributes being used may not be sufficiently informative about catchment-scale soil and vegetation contributions to hydrological behaviors. So, is it a problem of poorly encoded soils and vegetation information at the catchment scale, or is really the case that such soils and vegetation do not play as big a role in hydrological behavior as we might expect? It would be interesting to consider how this issue could be better investigated using the ML approach.

[17] In conclusion, I commend the authors for a very interesting and thought-provoking article, and I recommend the paper for publication after only minor revisions, in which the authors can chose to incorporate some of my review comments (or responses to them), or not, as they so choose. Best Regards Hoshin Gupta

References: Gupta HV, C Perrin, R Kumar, G Blöschl, M Clark, A Montanari and V Andressian (2014), Large-Sample Hydrology: A Need to Balance Depth With Breadth, special issue of HESS-ESD 'Predictions under change: water, earth, and biota in the anthropocene; Eds: M. Sivapalan, T. J. Troy, V. Srinivasan, A. Kleidon, D. Gerten, and A.

Montanari, Hydrology and Earth Systems Science, 18, 1–15, www.hydrol-earth-syst-sci.net/18/1/2014/ doi:10.5194/hess-18-1-2014

Schaefli B and HV Gupta (2007), Do Nash values have value?, Hydrological Processes, 21(15), 2075-2080, simultaneously published online as Invited Commentary in Hydrologic Processes (HP Today), Wiley InterScience, doi: 10.1002/hyp.6825

Bulygina N, and H Gupta (2011), Correcting the mathematical structure of a hydrological model via Bayesian data assimilation, Water Resources Research, 47, W05514, doi:10.1029/2010WR009614

Bulygina N and HV Gupta (2010), How Bayesian Data Assimilation Can be Used to Estimate the Mathematical Structure of a Model, Stochastic Environmental Research

and Risk Assessment, 24:925–937 DOI 10.1007/s00477-010-0387-y

Bulygina N, and HV Gupta (2009), Estimating the uncertain mathematical structure of a water balance model via Bayesian data assimilation, Water Resources Research, 45, special issue on 'Uncertainty Assessment in Surface and Subsurface Hydrology', W00B13, doi:10.1029/2007WR006749.

Gupta HV and GS Nearing (2014), Debates—The future of hydrological sciences: A (common) path forward? Using models and data to learn: A systems theoretic perspective on the future of hydrological science, Invited Commentary, Water Resources Research, 50, doi: 10.1002/2013WR015096

Nearing GS and HV Gupta (2015), The Quantity and Quality of Information in Hydrologic Models, Water Resources Research, 51, 524–538, doi:10.1002/2014WR015895

Please also note the supplement to this comment:
https://www.hydrol-earth-syst-sci-discuss.net/hess-2019-368/hess-2019-368-RC2-supplement.pdf

---

## Referee Comment (RC3) · Anonymous Referee #3 · 16 Sep 2019

Benchmarking LSTM

Overall, this paper stands at the forefront of hydrology. There are three aspects of the paper that I like. First, this work show state-of-the-art performance in terms of large-scale streamflow prediction accuracy. This would serve to push hydrologic science forward. Second, the authors implemented a novel LSTM structure to enable a static layer through which they could examine the impacts of different static catchment attributes. Third, they investigated network internal embeddings which is the first time in hydrology which I have seen, and provided some insights (not so perfect, as I would expand on later). These are all novel and I believe the paper should eventually be accepted.

Upon deeper examination I indeed found some issues related to potentially un-robust analysis, points of confusion and lack of clarity, need for more hydrologic insights, and somewhat superficial discussion in the exploration of embeddings. Some relevant citations are also missing. Thus I rate the manuscript a moderate revision. The comments below are not to cast the paper in a negative way, but they are in the hope of helping the authors improve the paper to a strong state before publication.

Major comments:

1. Hydrologic understanding: the discussion of the clustering and embeddings was, shall I say, not entirely satisfying. I liked the novelty of the visualization and the construct of the LSTM to enable this. It helped us understand a bit more about how LSTM works. However, I craved for a bit more hydrologic understanding. The discussion in section 3.4 was a bit sporadic and not so memorable. The take-home message appears to be "the EA-LSTM is able to learn complex interactions between catchment attributes that allows for grouping different basins". Stopping here does not help with the long-standing criticism of machine learning as a blackbox. I had hoped to gain some deeper hydrologic insights, e.g., why different basins were grouped together? What is the characteristic of each cluster and how are these clusters different from previous catchment clustering schemes, e.g., (Berghuijs et al., 2014; Carrillo et al., 2011; Fang & Shen, 2017; Sawicz et al., 2011; Toth, 2013; Troch et al., 2013)? To go deeper it may not need additional work, but more thoughts about the results.

2. More robustness: I'm afraid many of the attributes in Table 4 are correlated in space and it may be not very robust to draw conclusions from them especially for attributes that are not the highest ranking. For example, does geological permeability really stand position #9? Can we take it that permeability is the second important factor amongst non-climatic factors? This is somewhat surprising and is worth more discussion, but I'm afraid it might just be due to coincidence. To see so the authors could remove some basins (randomly or removing a spatial cluster) or attributes (as the factors tend to have interaction in these kinds of factor analysis) and train again and see how this table react to the perturbation.

3. Details for reproducibility: one of the selling points of the paper was the high performance. Hence it imperative that the results are reproducible. Are the transformations applied for input and output? How many layers of LSTM were used (in comparison with authors' HESS 2018 paper, this choice seemed ad hoc?)? How was the ranking for Table 4 done indeed? This was a local method, so what is the origin for perturbation?

4. Share more experience please: there are many choices which were unexplained, and the community would benefit from the authors providing more discussion of what worked and what did not during their experiments. How did other objective functions do? What if you don't do

ensemble averaging? How large are the impacts of hyperparameters, e.g., hidden layers and learning rates? These do not necessarily need figures and could be answered by a couple of sentences. Some minor points below are related to this.

5. The authors should also expand on why climatic factors showed up on top of table 4. It appears other static basin physical attributes were not important at all. Does this suggest catchment co-evolution? A potential indication of overfitting (to climatic factors that obviously vary), and more discussion is begging to be done here.

Minor points:

1. I'm at a loss to understand the opening statement about streamflow being an out-standing problem. At what point is this problem solved vs not solved? Is there a hard threshold? Did the present work solve this problem?
2. L73, "which part of the network are used for a given basin"—this sentence is difficult to interpret at this point. What does "used for" mean here.
3. L76, "similarly behaving". Is this referring streamflow responses or attributes? (only the former would be called a behavior, but this work didn't seem to include streamflow response in the clustering part)
4. L78, "embedding". This is a natural language processing jargon. Quite difficult for hydrologists to comprehend. I think it would be reader friendly if the authors spend two sentences explaining this word. My understanding is that embeddings are not just hidden layer activations, but a mapping of inputs to an ordered hidden space that has meanings. For example, the hidden layers of machine translation layers form an embedding. Each ranked item in the embedding in NLP can be related to a linguistic concept.
5. L117 "some amount of information" is fuzzy. Is it about catchment attributes or about streamflow responses? This is critically important as the two have very different meaning regarding what would be done. From reading the later parts, here you seem to refer to static attributes.
6. L122, regarding using static attributes as a constant array. It would be relevant to cite (Fang et al., 2017) which used this setup and was already distinguishing different landscapes using static attributes as inputs to LSTM. It occurs this paper should at least be mentioned in the present one.
7. L134-135. This is an interesting setup. It's worth mentioning that, from Eq 9 & 11, what was selected by the input gate were not only $x_d$ but also $h$ from the last step.
8. L158 – what happened when you used other loss functions?
9. L171 "25,000 km2" – is it really appropriate to model those with an area of 25,000 km2 the same as other smaller basins?
10. L194 – "favor of"
11. L222, regarding the ensemble averaging, readers deserve to know, how big is the spread? What if you don't take the average? Sometimes the ensemble mean gets a better R2 but it misses peaks.
12. It is unclear what "six different settings and eight different models" are.
13. L261. might be useful to say you extracted gradients from the learned network after training (correct?), as some readers are unfamiliar with how this is done. However, these gradients are time-step dependent.

14. Also, why is it called global sensitivity test? It is also local, around a origin for perturbation.
15. L264 better say "the average of absolute gradients across all basins and all time steps", and---- why absolute?
16. L267-268 "represent xxx into xxx"? the sentence does not make grammar sense. please fix. This is obviously an expansion of from 27 to 256. Why would this be really necessary?
17. Table 2. this value is indeed the highest I have seen. Good work!
18. L380. Why 447 basins now? What are missing?
19. L410 Unsure how this answers the question if the network just remembers. The logic is confusing.
20. L414, mean precipitation, etc --- aren' these supposed to be climatic inputs rather than static? (can we not let the network generalize it from the forcing data)?
21. Table 4. Echoing a major point raised above. What further conclusions can be drawn from the fact that climatic attributes take the most important positions? catchment Co-evolution theory?
22. L454 "before vs. after the transformation into the embedding layer". This is a good comparison, although later there didn't seem to be much comment on this comparison
23. UMAP—might be good to briefly explain what it does. Is it just PCA?
24. L479 Honestly, it's not that easy to see which cluster you are talking about. could use some annotation on the plot.
25. L489 I found this discussion, as a take-home message, to be somewhat superficial, and unsurprising. I'd appreciate somewhat more in-depth discussion about the hydrology.
26. Figure 11 these colors do not mean anything. It is a bit confusing. Why not use a at least partially consistent color scheme?
27. Figure 12 better annotate axes even if they don't mean much
28. L524. I am confused why this is called regional, as the LSTM was trained with all basins over CONUS. What would constitute a model that is not regional?
29. L529-530. It either goes against a belief or it does not. Can't go "somewhat against". And, the logic here is not quite clear. This paper is not about parameter identification. The fact that the network works does not imply that parameters can be identified. First the LSTM parameters cannot be interpreted. Second, even very different parameters could give you similar predictions.

---

## Referee Comment (RC4) · Anonymous Referee #3 · 16 Sep 2019

I forgot to attached the literature cited in my PDF file so here they are:

Berghuijs, W. R., Sivapalan, M., Woods, R. A., & Savenije, H. H. G. (2014). Patterns of similarity of seasonal water balances: A window into streamflow variability over a range of time scales. Water Resources Research, 50(7), 5638–5661. https://doi.org/10.1002/2014WR015692

Carrillo, G., Troch, P. A., Sivapalan, M., Wagener, T., Harman, C., & Sawicz, K. (2011). Catchment classification: hydrological analysis of catchment behavior through process-based modeling along a climate gradient. Hydrology and Earth System Sciences, 15(11), 3411–3430. https://doi.org/10.5194/hess-15-3411-2011

[Figure]

Fang, K., & Shen, C. (2017). Full-flow-regime storage-streamflow correlation patterns provide insights into hydrologic functioning over the continental US. Water Resources Research, 53(9), 8064–8083. https://doi.org/10.1002/2016WR020283

Fang, K., Shen, C., Kifer, D., & Yang, X. (2017). Prolongation of SMAP to Spatio-temporally Seamless Coverage of Continental US Using a Deep Learning Neural Network. Geophysical Research Letters, 44, 11030–11039. https://doi.org/10.1002/2017GL075619

Sawicz, K., Wagener, T., Sivapalan, M., Troch, P. A., & Carrillo, G. (2011). Catchment classification: empirical analysis of hydrologic similarity based on catchment function in the eastern USA. Hydrology and Earth System Sciences, 15(9), 2895–2911. https://doi.org/10.5194/hess-15-2895-2011

Toth, E. (2013). Catchment classification based on characterisation of streamflow and precipitation time series. Hydrology and Earth System Sciences, 17(3), 1149–1159. https://doi.org/10.5194/hess-17-1149-2013

Troch, P. A., Carrillo, G., Sivapalan, M., Wagener, T., & Sawicz, K. (2013). Climate-vegetation-soil interactions and long-term hydrologic partitioning: signatures of catchment co-evolution. Hydrology and Earth System Sciences, 17(6), 2209–2217. https://doi.org/10.5194/hess-17-2209-2013

---

## Author Response (AR1)

Comments/Text of **Anonymous Referee 1 (AR1)** posted in blue, and our answers in black with old passages in red and new passages in green.

This very interesting paper of Kratzert, et al. compares the quality of the predictions of various hydrological models with three variants of the Long Short-Term Memory(LSTM) deep learning network. One of these variants, the novel EA-LSTM, is trained using meteorological data and catchment similarities as an additional input and is analysed in detail highlighting the superiority of such a network. In general the paper is very well written and it is worth to be published after some minor changes. Some comments:

1. Maybe you could explain better the differences of the analysis of the single model and the ensemble mean approach. On page 13, lines 317-320 you write: "To assess statistical significance for single models, the mean basin performance (e.g. mean NSE per basin and across all seeds) between two different model settings was compared between different model configurations." What's the difference between model settings and configuration? If I understood it correctly the difference in the verification of the single models and the ensemble mean is: Single model: From 8 ensemble model runs, you get 8 different predictions and you calculate the verification measures (e.g. NS values) for each of it and take the average (+/- Std? in Table 2); whereas in the Ensemble mean approach, for example this measure is calculated taking the mean of the 8 predictions?

Thank you. We will rewrite this section of the description of methods (lines 317ff) to more clearly describe the ensemble approach and how the statistics of the single model are calculated.

Old passage:
To assess statistical significance for single models, the mean basin performance (e.g. mean NSE per basin and across all seeds) between two different model settings was compared between different model configurations. To assess statistical significance for ensemble means, the mean basin performance of the ensemble mean was compared between different model configuration.

New passage:
To assess statistical significance for single models, we first calculated the mean basin performance, i.e. the mean NSE per basin across the 8 repetitions. The so derived mean basin performance was then used for the test of significance. To assess statistical significance for ensemble means, the ensemble prediction (i.e. the mean discharge prediction of the 8 model repetitions) was used to compare between different model approaches.

2. For clarity reasons I would not include the single model outcome in Figure 3, because this a random outcome and would look different for each ensemble run.

We don't agree with the reviewer on this point because we think it helps to underscore the potential of ensembling. As shown in Table 2 and Table 3 in the manuscript and the figure below, the overall CDF of the single models does have little variation between random seeds, especially in comparison to the benefit of ensembling. Therefore, we would like to keep these curves in the figure, since it helps to visualize the ensembling benefit and show by how much the CDF can be improved.

[Figure]

3. Nice to have the significance reported, which is most often not shown. Although the precision of these p-values is extremely high and the differences are probably rather neglectable caused by noise.

We agree with the reviewer that reporting significance measures is important.

4. Regarding the modified NSE. Wouldn't it be easier to normalize the streamflow data (e.g. using the BoxCox transformation)? So you don't have to event a new measureand adding a constant in order to achieve stable results.

The loss function we use isn't really new. It's just the basin-average NSE. The reason this is a little unusual in Hydrology is because we rarely (if ever) calibrate a single model to multiple basins. But this is standard practice in machine learning - where the overall loss function is the average loss over many samples. The only alternative is a single loss function calculated over concatenated data from multiple samples, which doesn't work well (or at least not as

straightforward) for stochastic gradient descent, which randomizes and sub-samples the training samples.

Moreover, the reason we did not transform the data before training is because this affects model performance (besides normalizing to zero mean, unit variance). For example, an exponential-type transform like Box-Cox will generally under-emphasize peak flows (if the box-cox exponent is <1). We did try calibrating to log-transformed data, in an effort to normalize the streamflow data, but this does not work as well as using the natural streamflow data. The goal is to train to the non-transformed target data, but use a loss function that does not overemphasize any particular training sample (i.e., any particular or individual basin does not have out-sized influence on the training procedure).

5.  Looking at the results, I would conclude that the EA-LSTM is very interesting fort his analysis, but for practical applications the LSTM with the coupled meteo data and catchment attributes is even more efficient and is less complex. That's why I would like to see the results of this model also in Figure 4, 5 and Table 3.

We will modify Figure 4, Figure 5 and Table 3 to include the results of the standard LSTM.

New passage added to the beginning of Sect. 3.2:
In this section, we concentrate on benchmarking the EA-LSTM, however for the sake of completeness, we added the results of the LSTM with static inputs to all figures and tables.

6.  Are the catchment attributes kept static for all days of the year?  For example the monthly mean of leaf area index could be easily varied depending on the month of the year?

Yes, the catchment attributes are static in this study. We are currently working on making these dynamic, including vegetation and climate indexes, soil moisture, snow cover from remote sensing, etc. This is however not a trivial extension and we do believe that the idea is worth being studied separately. Furthermore, for the sake of repeatability, we wanted to stick entirely to the CAMELS data set, which only includes static catchment attributes. Right now, in this paper, we are using long-term catchment attributes as indicators of differences between catchments (regional heterogeneity among catchments), not for assessing nonstationary catchment behaviors.

7. I would suggest to delete the UMAP analysis, since the method is not explained and the results are a bit confusing.

On this point we don't agree with the reviewer and would like to keep this section. We see this as an interesting addition to the introduction of the new EA-LSTM and the benchmarking results. Specifically, we are using this analysis to illustrate the fact that the embedding layer (our static LSTM input gate) can 'learn' about catchment diversity in a physically meaningful way. This is a (fairly simple) form of explainable AI, and one of the goals of this paper is to work toward that

larger objective. The analysis of the embedding layer is important as an example of this larger purpose, and the UMAP analysis in particular is necessary for a reduced-dimension (i.e., graphical) analysis.

Although our paper is mainly intended as a modeling paper and we see the introduction of the EA-LSTM and benchmarking against various hydrological models as our main contributions, we think keeping Section 3.4 has the following benefits:
- It provides at least some feeling about what happened in the embedding layer of the input gate in the EA-LSTM (namely grouping of basins that match our expert knowledge).
- As such, it helps in our opinion to gain trust into LSTMs which are widely considered as black-box model.
- It shows possible analysis that are possible with the EA-LSTM in general and potentially open the door for many follow-up studies in the future. Such studies could either concentrate more on the interpretability of LSTM-based models or try to extract new hydrological understanding from the learned groupings.

We will extend the manuscript to include a short description of the UMAP method, however, would avoid a lengthy discussion on the method and point the interested reader to the official UMAP publication. In our manuscript it is simply take as (state-of-the-art) dimension reduction technique.

New passage describing UMAP:
Finally, we reduced the dimension of the input gate embedding layer (from R^256 to R^2) so as to be able to visualize dominant features in the input embedding. To do this we use a dimension reduction algorithm, called UMAP (McInnes et al., 2018) for "Uniform Manifold Approximation and Projection for Dimension Reduction". UMAP is based on neighbour graphs (while e.g., principle component analysis is based on matrix factorization), and it uses ideas from topological data analysis and manifold learning techniques to guarantee that information from the high dimensional space is preserved in the reduced space. For further details we refer the reader to the original publication by McInnes et al. (2018).

Comments/Text of **Hoshin Gupta (Reviewer 2)** posted in blue, and our answers in black with old passages in red and new passages in green.

([1-11] present a thoughtful summary of our manuscript)

[12] I believe that this paper represents a very significant contribution to the Earth System literature related to the development of Dynamical Environmental Systems Models (DESMs). I have alluded to some of the problems associated with the conventional CM approach in paragraphs [2-6] above. In this regard, there has been increasing community interest in the use of both "large sample" data sets and the use of "model-structural-correction-via-data-assimilation" (learning from data) to extract better understanding about the structure and functioning of hydrological systems, such as catchments.

[13] This paper bridges the challenges of learning from large sample data sets and learning how catchments structures/behaviors can differ at local to regional scales in a very meaningful way. While not addressing the problem of prediction in ungaged basins directly, the ability of the EA-LSTM to learn from and characterize differences in catchment functioning encoded in catchment attribute data is highly significant, and it would seem that a natural next step would be for the authors to demonstrate that potential by running experiments that seek to demonstrate that predictive ability learned from gaged locations can be transferred to ungaged locations. I look forward to reading more about this in the future.

We do have a short paper on this topic in WRR, that this reviewer is also currently reviewing. That paper, however, does not explore how the catchment-aware embedding presented here as an adaptation of the LSTM architecture helps in the PUB setting. This is for future work.

[14] As such, I have only a few suggestions to offer the authors. The first is that the current title "Benchmarking a Catchment-Aware Long Short-Term Memory Network (LSTM) for Large-Scale Hydrological Modeling" presents a rather technical front to what is arguably (in my opinion) a much more significant piece of work. I therefore offer up the possibility for the authors to consider that the introduction and discussion/conclusions sections be somewhat revamped/broadened to reflect the perspectives offered in my above summary of the paper. As indicated, I do think this paper is really more about the interesting challenges of learning and characterizing (via dynamical systems models) the "behavior and functioning" of hydrological systems at the catchment scale in such a manner that both universal (fundamentally hydrological) principles, and local-to-regional scale uniquenesses of such systems can be learned by accessing the patterns of information encoded in large sample data sets (Gupta et al 2014). In this regard the title could also then be generalized to reflect the nature of the conversation about "Learning Universal, Regional and Local Hydrological Behaviors via Machine-Learning applied to Large Sample data Sets". Or this more general discussion could be saved for a future publication.

Thank you for the suggestion about broadening the scope implied by the title. We will take the suggestion to change the title and update the discussion accordingly. However we don't feel confident enough to state that we already have a "universal" model, but rather that this work is a step in that direction. Thus, we would change the title to "*Towards Learning Universal, Regional, and Local Hydrological Behaviors via Machine-Learning Applied to Large-Sample Datasets*". We Furthermore adapted the first paragraphs of the discussion as follows:

New passage (beginning of discussion section):
The EA-LSTM is an example of what Razavi and Coulibaly (2013) called a model-independent method for regional modeling. We cited Besaw et al. (2010) as an earlier example this type of approach, since they used classical feed-forward neural networks. In our case, the EA-LSTM achieved state-of-the-art results, outperforming multiple locally- and regionally-calibrated benchmark models. These benchmarking results are arguably a pivotal part of this paper.
The results of the experiments described above demonstrate that a single 'universal' deep learning model can learn both regionally-consistent and location-specific hydrologic behaviors. The innovation in this study – besides benchmarking the LSTM family of rainfall-runoff models – was to add a static embedding layer in the form of our EA-LSTM. This model offered similar performance as compared with a conventional LSTM (Sect. 3.1) but offers a level of interpretabiltiy about how the model learns to differentiate aspects of complex catchment-specific behaviors (Sect. 3.3 and Sect. 3.4). In a certain sense, this is similar to the aforementioned MPR approach, which links its model parameters to the given spatial characteristics (in a non-linear way, by using transfer functions), but has a fixed model structure to work with. In comparison, our EA-LSTM links catchment characteristics to the dynamics of specific sites and learns the overall model from the combined data of all catchments. Again, the critical take-away, in our opinion, is that the EA-LSTM learns a single model from large catchment data sets in a way that explicitly incorporates local (catchment) similarities and differences

[15] The second is that while the basin-average NSE* loss function does seem to serve the immediate needs of this study, I think that the ML-approach (and more generally hydrological learning from catchment data sets) can benefit from a more thoughtful approach to the problem of model performance metrics. In particular, the use of the observed output data "mean" as a benchmark for constructing the NSE itself, and the use of the output data variance to "normalize" across catchments to obtain somewhat comparable metric values to be averaged (or otherwise summarized in some statistical manner) seems, to me, problematic. In this regard, I think an Information Theoretic approach might ultimately prove to be more meaningful. I point out that the value of the metric, when used as the basis for assessing across different catchment locations, would be much enhanced if it somehow recognized the relative differences in complexity/difficulty associated with modeling the dynamical input-state-output behaviors at different locations (due to climatic, geological, and other factors). As discussed by Schaefli and Gupta (2007), the problem is at least partly one of appropriate benchmarking in order to make metric values meaningfully comparable. Some types of catchments (such as humid ones perhaps) are relatively easy to model to the level of obtaining high performance (e.g. NSE)

values, while others (such as arid ones perhaps) are much more difficult to model … potentially requiring more complex model structures, more data, and perhaps better data quality. Since the challenge here is learning hydrological principles from the data, and some catchment systems are easier to characterize using simpler model structures, it would seem prudent to figure out how to account for this knowledge in the designs of our learning systems, which includes the metrics used as the filter through which information is being extracted.

We absolutely agree that it will be critical, going forward, to understand carefully and in detail how different loss functions affect the training of deep learning Hydrology models. We've done some work with this - trying to emphasize peak and low flows, working with probabilistic loss functions, etc. None of that work is mature enough to publish at this point. This has been a much bigger challenge that we perhaps originally expected, and we appreciate the reviewer's advice. Hopefully we will have something more meaningful or helpful to say about this in a future publication.

[16] Finally, I think that the aforementioned issue may also relate to the fact that certain catchment attributes tend to be dominant indicators of differences in catchment behaviors, while others seem to show "lower importance" (sensitivity). It is been well known that "climate" (and one would reasonably expect also "topography") is the dominant indicator of catchment similarity, but this does not really help us to understand what structural differences in catchments drive differences in their behaviors. The finding that soil and vegetation characteristics are low on the "importance" list is interesting, as it suggests that the existing catchment attributes being used may not be sufficiently informative about catchment-scale soil and vegetation contributions to hydrological behaviors. So, is it a problem of poorly encoded soils and vegetation information at the catchment scale, or is really the case that such soils and vegetation do not play as big a role in hydrological behavior as we might expect? It would be interesting to consider how this issue could be better investigated using the ML approach.

First we would like to clarify that we do not say that soil and vegetation indices are not "important" but just that climatic and topographic attributes are *more* important. As the reviewer mentions, this follows hydrological literature and also the intuition of the reviewer. In our case, this could potentially result in two basins with similar climatic and topographic attributes that are distinguished primarily by their soil/vegetation properties. However, in the larger context it is the set of climate and topographic attributes that "separates" the most basins and thus the larger sensitivity/importance.

Regarding the question about what we might be able to learn from these results for hydrological modeling. This seems in-line with the well-known idea that the first-order trends in most models are Budyko-type effects (i.e., climate related). This is not especially new, but also encouraging that the LSTM behaves as expected. One thing we might take away from this is due to the comparison with MPR regionalization. MPR uses topographic, geologic, soil and land use attributes as inputs. We show that our regionalization approach outperforms the two MPR calibrated models (VIC and mHM). This could indicate that (the conventional use of) MPR might

be improved by including climatic attributes in the regionalization scheme.. I guess the question is about the extent to which it is meaningful for a model to 'react' directly to climate indexes rather than just to meteorological forcings. The regressions in typical regionalization strategies could use climate indexes as regressors or inputs.

Another point regarding the relative unimportance of geological and vegetation features indicated in our sensitivity analysis is that probably catchment averaged soil properties, as well as vegetation indices contain too much noise compared to the relatively noise-free climatic and topographic attributes. This is what is probably meant by the reviewer with "poorly encoded" information of those features, in which case we would agree. We don't, however, prove this in the paper - all we show in the paper is that there is enough information in the indexes to at least help with catchment differentiation, and that traditional approaches do not utilize all of this information.

Comments/Text of **Anonymous Referee 3 (AR3)** posted in blue, and our answers in black with old passages in red and new passages in green.

Benchmarking LSTM

Overall, this paper stands at the forefront of hydrology. There are three aspects of the paper that I like. First, this work show state-of-the-art performance in terms of large-scale streamflow prediction accuracy. This would serve to push hydrologic science forward. Second, the authors implemented a novel LSTM structure to enable a static layer through which they could examine the impacts of different static catchment attributes. Third, they investigated network internal embeddings which is the first time in hydrology which I have seen, and provided some insights (not so perfect, as I would expand on later). These are all novel and I believe the paper should eventually be accepted.

Upon deeper examination I indeed found some issues related to potentially un-robust analysis, points of confusion and lack of clarity, need for more hydrologic insights, and somewhat superficial discussion in the exploration of embeddings. Some relevant citations are also missing. Thus I rate the manuscript a moderate revision. The comments below are not to cast the paper in a negative way, but they are in the hope of helping the authors improve the paper to a strong state before publication.

Major comments:

1. Hydrologic understanding: the discussion of the clustering and embeddings was, shall I say, not entirely satisfying. I liked the novelty of the visualization and the construct of the LSTM to enable this. It helped us understand a bit more about how LSTM works. However, I craved for a bit more hydrologic understanding. The discussion in section 3.4 was a bit sporadic and not so memorable. The take-home message appears to be "the EA-LSTM is able to learn complex interactions between catchment attributes that allows for grouping different basins". Stopping here does not help with the long-standing criticism of machine learning as a blackbox. I had hoped to gain some deeper hydrologic insights, e.g., why different basins were grouped together? What is the characteristic of each cluster and how are these clusters different from previous catchment clustering schemes, e.g., (Berghuijs et al., 2014; Carrillo et al., 2011; Fang & Shen, 2017; Sawicz et al., 2011; Toth, 2013; Troch et al., 2013)? To go deeper it may not need additional work, but more thoughts about the results.

To avoid misunderstandings, we would like to clarify that we see the main contribution of this paper as i) demonstrating of the LSTM-based modeling approach for large-scale hydrological modeling in general (building upon the results of Kratzert et al., 2018) ii) introducing the EA-LSTM and iii) benchmarking vs. a large set of well-established hydrological models.

With this premise we would like to address the comment regarding  Section 3.4: Within our EA-LSTM, we include an embedding layer (the static LSTM input gate), that can 'learn' about catchment diversity purely from discharge data. Analyzing the (physically) meaningfulness of the learned embedding can be seen as a (fairly simple) form of explainable AI. Although, as said above, our paper is mainly intended as a modeling paper, we think that Section 3.4 has the following benefits (copied from our answers to Reviewer #1):

- The blackbox i somewhat opened. The section provides at least some intuition about what happened in the embedding layer of the input gate in the EA-LSTM (namely grouping of basins in a way that matches expectations).
- It provides an example of the kind of analysis that are possible with the EA-LSTM in general and potentially opens the door for many follow-up studies in the future. Such studies could either concentrate more on the interpretability of LSTM-based models or try to extract new hydrological understanding from the learned groupings.

Given the above mentioned scope and the benefits of the section, we would like to avoid extending the hydrological interpretation of Section 3.4. Especially, because here we are not analyzing the model performance, but rather just examine the intrinsic properties of the model. Additionally, as the reviewer herself/himself cites, doing a full-fledged cluster analysis is the work of many individual publications themselves and would clearly be out of scope here.

2. More robustness: I'm afraid many of the attributes in Table 4 are correlated in space and it may be not very robust to draw conclusions from them especially for attributes that are not the highest ranking. For example, does geological permeability really stand position #9? Can we take it that permeability is the second important factor amongst non-climatic factors? This is somewhat surprising and is worth more discussion, but I'm afraid it might just be due to coincidence. To see so the authors could remove some basins (randomly or removing a spatial cluster) or attributes (as the factors tend to have interaction in these kinds of factor analysis) and train again and see how this table react to the perturbation.

First off, we would like to state that any spatial correlation in physical catchment features is real information that can and should be leveraged by regional models. We even explicitly did not include latitude/longitude inputs to our model in this study so that only real, physically-based information is leveraged directly by the EA-LSTM.
This is discussed, for example, by Addor et al (2018), and our findings are in line with the results of said publication. The results may thus be less surprising than indicated in this comment. In this context we would also like to mention that we did an independent robustness analysis by perturbing the features with gaussian noise (see L395ff), which shows the reliance (and robustness) of the model with respect to changes of the features.

We do however agree that the results do not form a particularly strong ranking. Regarding this point, it is important for us to emphasize that in the original contribution did not claim anywhere that the absolute rank of any particular feature has a meaning. This was a model sensitivity

analysis, which is common for modeling studies. The only conclusions that we drew from this sensitivity analysis are:

- "..*the most sensitive catchment attributes are topological features [...] and climate indices [...].*" (L 422f)
- "*Certain groups of catchment attributes did not typically provide much additional information. These include vegetation indices [...], as well as the annual vegetation differences. Most soil features were at the lower end of the feature ranking*" (L 423ff)

These seem to be valid conclusions of a sensitivity analysis like this.

That said, our experiments suggest that the obtained qualitative ranking of the feature groups (like climatic, topological, soil and vegetation) is rather robust. To strengthen upon this statement, we added below the results of the same analysis for all 8 repetitions of the same model settings (the EA-LSTM optimized with the basin average NSE) as used in Table 4. As we can see from these tables, the qualitative ranking of these feature groups remained similar. We hope that the reader does focus on exact rankings or exact sensitivity values of any particular feature but rather on the overall image of Table 4 - which is why we grouped these into categories in the first place.

We also agree with the reviewer that this might not be clear from the way the manuscript is currently written, and thus the results could be questioned as being a *coincidence.* We will therefore update the manuscript to clarify regarding this point.

Newly added passage:
Note that the results between the 8 model repetitions (not shown here) vary slightly in terms of sensitivity values and ranks. However, the quantitative ranking is robust between all 8 repetitions, meaning that climate indices (e.g. aridity and mean precipitation) and topological features (e.g. catchment area and mean catchment elevation) are always ranked highest, while soil and vegetation features are of less importance and are ranked lower. It is worth noting that our rankings qualitatively agree with much of the analysis by Addor et al. (2018).

```
p_mean               0.682248    p_mean               0.737930
aridity              0.564276    elev_mean            0.620351
area_gages2          0.504591    area_gages2          0.520789
elev_mean            0.459893    frac_snow            0.481682
high_prec_dur        0.406671    clay_frac            0.471752
frac_snow            0.405564    aridity              0.407523
high_prec_freq       0.382006    gvf_max              0.335463
slope_mean           0.370855    geol_permeability    0.296405
geol_permeability    0.352949    soil_depth_pelletier 0.291952
carbonate_rocks_frac 0.339022    pet_mean             0.290072
clay_frac            0.330383    slope_mean           0.282145
pet_mean             0.310769    low_prec_freq        0.280580
low_prec_freq        0.299585    soil_depth_statsgo   0.274308
soil_depth_pelletier 0.273934    soil_conductivity    0.274178
p_seasonality        0.272786    silt_frac            0.259220
frac_forest          0.267421    high_prec_dur        0.259150
sand_frac            0.255156    p_seasonality        0.250047
soil_conductivity    0.243641    low_prec_dur         0.248930
low_prec_dur         0.219104    sand_frac            0.243023
gvf_max              0.213809    carbonate_rocks_frac 0.235574
gvf_diff             0.212412    frac_forest          0.232315
lai_diff             0.208096    gvf_diff             0.222452
soil_porosity        0.194036    high_prec_freq       0.202322
soil_depth_statsgo   0.191936    lai_diff             0.192319
lai_max              0.190274    lai_max              0.168555
silt_frac            0.183365    soil_porosity        0.167466
max_water_content    0.158722    max_water_content    0.142825

                                 elev_mean            0.687521
elev_mean            0.592299    p_mean               0.675504
p_mean               0.576201    aridity              0.524162
aridity              0.506481    low_prec_dur         0.427454
area_gages2          0.474934    soil_depth_pelletier 0.403478
frac_snow            0.447427    clay_frac            0.401946
clay_frac            0.443380    frac_snow            0.396246
carbonate_rocks_frac 0.432280    area_gages2          0.384397
slope_mean           0.400347    high_prec_dur        0.383352
geol_permeability    0.368472    slope_mean           0.378002
pet_mean             0.368429    gvf_max              0.345176
soil_depth_pelletier 0.342860    low_prec_freq        0.342597
gvf_max              0.329133    pet_mean             0.327042
sand_frac            0.314407    geol_permeability    0.323855
high_prec_freq       0.301476    p_seasonality        0.322263
soil_conductivity    0.295581    frac_forest          0.320437
high_prec_dur        0.279304    silt_frac            0.292820
gvf_diff             0.279032    high_prec_freq       0.273888
p_seasonality        0.276549    max_water_content    0.245695
silt_frac            0.267486    soil_depth_statsgo   0.245633
low_prec_freq        0.233236    sand_frac            0.203782
soil_porosity        0.212450    soil_conductivity    0.201373
soil_depth_statsgo   0.212345    gvf_diff             0.195265
frac_forest          0.193351    carbonate_rocks_frac 0.188165
lai_max              0.192644    lai_diff             0.173871
lai_diff             0.185409    lai_max              0.141322
max_water_content    0.171502    soil_porosity        0.113825
low_prec_dur         0.167153    dtype: float64
```

| | | | |
|---|---|---|---|
| p_mean | 0.683777 | elev_mean | 0.614620 |
| elev_mean | 0.535805 | p_mean | 0.600607 |
| aridity | 0.474985 | frac_snow | 0.515789 |
| area_gages2 | 0.473146 | aridity | 0.506338 |
| frac_snow | 0.430077 | area_gages2 | 0.439643 |
| high_prec_freq | 0.429197 | soil_depth_pelletier | 0.366733 |
| slope_mean | 0.406997 | slope_mean | 0.346615 |
| soil_depth_pelletier | 0.387183 | clay_frac | 0.343064 |
| carbonate_rocks_frac | 0.375872 | carbonate_rocks_frac | 0.329496 |
| clay_frac | 0.357481 | gvf_max | 0.318784 |
| geol_permeability | 0.344788 | high_prec_freq | 0.314248 |
| gvf_max | 0.327458 | p_seasonality | 0.309494 |
| gvf_diff | 0.324827 | low_prec_freq | 0.305372 |
| pet_mean | 0.320391 | geol_permeability | 0.285597 |
| low_prec_freq | 0.310324 | sand_frac | 0.285117 |
| p_seasonality | 0.291569 | high_prec_dur | 0.272177 |
| high_prec_dur | 0.273754 | low_prec_dur | 0.235999 |
| silt_frac | 0.272043 | pet_mean | 0.231612 |
| low_prec_dur | 0.241519 | silt_frac | 0.230806 |
| soil_depth_statsgo | 0.226876 | gvf_diff | 0.225315 |
| max_water_content | 0.219675 | soil_conductivity | 0.222055 |
| sand_frac | 0.215917 | frac_forest | 0.194465 |
| soil_conductivity | 0.214915 | soil_depth_statsgo | 0.189208 |
| frac_forest | 0.196826 | soil_porosity | 0.177579 |
| soil_porosity | 0.189546 | lai_diff | 0.166245 |
| lai_max | 0.173699 | lai_max | 0.157240 |
| lai_diff | 0.155421 | max_water_content | 0.143066 |

| | | | |
|---|---|---|---|
| elev_mean | 0.624811 | p_mean | 0.690424 |
| p_mean | 0.607064 | elev_mean | 0.563552 |
| aridity | 0.483825 | frac_snow | 0.557419 |
| area_gages2 | 0.437059 | aridity | 0.513616 |
| p_seasonality | 0.421215 | area_gages2 | 0.464579 |
| slope_mean | 0.390884 | high_prec_freq | 0.374508 |
| frac_snow | 0.390787 | high_prec_dur | 0.359950 |
| high_prec_freq | 0.382907 | gvf_diff | 0.333755 |
| high_prec_dur | 0.349156 | soil_depth_pelletier | 0.325056 |
| gvf_max | 0.349014 | geol_permeability | 0.324826 |
| geol_permeability | 0.335745 | pet_mean | 0.324743 |
| soil_depth_pelletier | 0.323830 | clay_frac | 0.322207 |
| carbonate_rocks_frac | 0.309022 | slope_mean | 0.317398 |
| gvf_diff | 0.288544 | gvf_max | 0.311988 |
| pet_mean | 0.287186 | sand_frac | 0.305872 |
| clay_frac | 0.284030 | low_prec_dur | 0.281842 |
| low_prec_freq | 0.245233 | p_seasonality | 0.278623 |
| frac_forest | 0.224514 | silt_frac | 0.271964 |
| soil_conductivity | 0.219839 | carbonate_rocks_frac | 0.267376 |
| silt_frac | 0.212018 | soil_conductivity | 0.259286 |
| low_prec_dur | 0.208801 | low_prec_freq | 0.237494 |
| sand_frac | 0.183467 | lai_diff | 0.178141 |
| soil_depth_statsgo | 0.180952 | frac_forest | 0.177963 |
| lai_diff | 0.178751 | lai_max | 0.177861 |
| max_water_content | 0.166533 | soil_porosity | 0.176054 |
| soil_porosity | 0.146432 | soil_depth_statsgo | 0.158091 |
| lai_max | 0.145553 | max_water_content | 0.127484 |

3. Details for reproducibility: one of the selling points of the paper was the high performance. Hence it imperative that the results are reproducible. Are the transformations applied for input and output? How many layers of LSTM were used (in comparison with authors' HESS 2018 paper, this choice seemed ad hoc?)? How was the ranking for Table 4 done indeed? This was a local method, so what is the origin for perturbation?

All information demanded by the reviewer are already reported in the manuscript:

- "Are the transformations applied for input and output?" L 247 "*All input features (both static and dynamic) were standardized (zero mean, unit variance) before training*". However, we agree that such an information should probably be placed in the data section (Section 2.4) and will update the manuscript accordingly.
- "How many layers of LSTM were used [...]?" The number of LSTM layers (one LSTM layer) is specified alongside the other network details in the Appendix B (L 562).
- "How was the ranking for Table 4 done[...]? " The details on how to derive the feature ranking is explained exhaustively in Section 2.6.2 "Robustness and Feature Ranking". Concretely, regarding the ranking of Table 4: "*Further, since we predict one time step of discharge at the time, we obtain this sensitivity measure for each static input for each day in the validation period. A global sensitivity measure for each basin and each feature is then derived from taking the average absolute gradient (Saltelli et al., 2004).*" and then L. 420f "*Table 4 provides an overall ranking of dominant sensitivities. These were derived by normalizing the sensitivity measures per basin to the range (0,1) and then calculating the overall mean across all features*"
- "... what is the origin for perturbation?" We used the optimized parameters as starting value. There might be some confusion here. We do not solve the gradient computation by numerical approximation, but rather calculate the gradients analytically through backpropagation. So if at all, the true values for the static catchment attributes can be seen as the origin of perturbation. In the original manuscript this is explained in 2.6.2 "Robustness and Feature Ranking" L.251f.

4. Share more experience please: there are many choices which were unexplained, and the community would benefit from the authors providing more discussion of what worked and what did not during their experiments. How did other objective functions do? What if you don't do ensemble averaging? How large are the impacts of hyperparameters, e.g., hidden layers and learning rates? These do not necessarily need figures and could be answered by a couple of sentences. Some minor points below are related to this.

Sadly, we do not know how we can do this. We tried to provide as much information as possible. And, to our knowledge, no choices in our network architecture or training procedure remained unexplained. Appendix B explains the hyperparameter search settings. We did not experiment with different learning rates and can't share any experiences on this question.

Furthermore, we did not test any other objective functions than the two reported in this paper. Hyperparameter search was performed using MSE (the machine learning community standard for regression tasks).

The only thing that comes to mind is that we did not report the results of all considered configurations and if wished we can update the Appendix B accordingly with a short description. As a short summary: The median model performance (across the basins) remains more or less stable between most configurations, while the most variance can be observed in the mean NSE. Two layers did not provide any meaningful improvement, that would justify the additional computational cost. However, our hyperparameter search was not exhaustive and at no point in the manuscript we claim to have found the best possible architecture for this task.

5. The authors should also expand on why climatic factors showed up on top of table 4. It appears other static basin physical attributes were not important at all. Does this suggest catchment co-evolution? A potential indication of overfitting (to climatic factors that obviously vary), and more discussion is begging to be done here.

The climatic factors show up on the top of the table, since through the method of Morris they have the highest gradient. We don't know of any experiment that would tell us *why* climate factors appear there (i.e. why they have the highest gradient), except hydrological intuition. (This is not different than any sensitivity analysis for any type of hydrologic model - sensitivity analyses do not answer questions about 'why' certain features are more sensitive) As such, these results in isolation do not suggest catchment co-evolution. They tell us that the model uses certain features more heavily than others. However, these findings are also in line with the results reported by Addor et al. (2018), as we state in L 428 "*It is worth noting that our rankings qualitatively agree with much of the analysis by Addor et al. (2018).*"

Also, this table doesn't suggest that physical attributes are unimportant, just that they are not as important as climate features. Again, this agrees with previous literature, as cited. This intuition that climate-related factors are the dominant drivers of hydrological systems, for example, models are often tested in terms of their ability to predict departures from the Budyko curve. We therefore do not see any indication of overfitting from this analysis.

Minor points:

1. I'm at a loss to understand the opening statement about streamflow being an out-standing problem. At what point is this problem solved vs not solved? Is there a hard threshold? Did the present work solve this problem?

To clarify: The sentence in question reads: "*Regional rainfall-runoff modeling is an old but still mostly out-standing problem in Hydrological Sciences*". Here, *out-standing* is referring to *regional* modeling, not to streamflow modeling in general. There is no hard threshold to determine when a problem like this is solved (and we believe that the sentence does not imply

that either; as a matter of fact we added the word "mostly" to avoid such a conclusion). However, the benchmarking in our paper with state-of-the-art regionalization methods and the fact that the proposed LSTM-based modeling approach significantly (and by far margins) outperforms these models, suggest that there is (or at least was) still significant room to improve how the community addresses this problem.

We believe that most readers will not be puzzled by the provided formulation and will therefore leave it unchanged.

2. L73, "which part of the network are used for a given basin"—this sentence is difficult to interpret at this point. What does "used for" mean here.

We added some clarity to this sentence: "*Concretely, we propose an adaption of the LSTM where catchment attributes explicitly control which parts of the LSTM state space are used for a given basin*"

Old passage:
Concretely, we propose an adaption of the LSTM where catchment attributes explicitly control which parts of the network are used for a given basin.

New passage:
Concretely, we propose an adaption of the LSTM where catchment attributes explicitly control which parts of the LSTM state space are used for a given basin.

3. L76, "similarly behaving". Is this referring streamflow responses or attributes? (only the former would be called a behavior, but this work didn't seem to include streamflow response in the clustering part)

"Behavior" here refers to the similarity in the rainfall-runoff dynamics, as suggested by the reviewer. This is also stated implicitly in the two sentences directly preceding the one in question (L74f) "*...it can learn how to combine different parts of the network to simulate different types of rainfall-runoff behaviors. In principle, the approach explicitly allows for sharing parts of the networks for similarly behaving basins...*"

4. L78, "embedding". This is a natural language processing jargon. Quite difficult for hydrologists to comprehend. I think it would be reader friendly if the authors spend two sentences explaining this word. My understanding is that embeddings are not just hidden layer activations, but a mapping of inputs to an ordered hidden space that has meanings. For example, the hidden layers of machine translation layers form an embedding. Each ranked item in the embedding in NLP can be related to a linguistic concept.

Historically, "embedding" is not a term from the field of natural language processing, but rather a general mathematical concept. Maybe the reviewer is confusing this term with "word embeddings", which is a term-of-art from natural language processing, but is not what we are

referring to. More importantly, L. 77f defines the term embedding exactly: "*..our adaptation provides a mapping from catchment attribute space into a learned, high-dimensional space, i.e. a so-called embedding*".

5. L117 "some amount of information" is fuzzy. Is it about catchment attributes or about streamflow responses? This is critically important as the two have very different meaning regarding what would be done. From reading the later parts, here you seem to refer to static Attributes.

We changed the previous sentence to: "*..our objective is to build a network that learns to extract information that is relevant to rainfall-runoff behaviors from observable catchment attributes.*" so that the context is hopefully clearer.

Old passage:
To reiterate from the introduction, our objective is to build a network that learns catchment similarities directly from rainfall-runoff data in multiple basins.

New passage:
To reiterate from the introduction, our objective is to build a network that learns to extract information that is relevant to rainfall-runoff behaviors from observable catchment attributes.

6. L122, regarding using static attributes as a constant array. It would be relevant to cite (Fang et al., 2017) which used this setup and was already distinguishing different landscapes using static attributes as inputs to LSTM. It occurs this paper should at least be mentioned in the present one.

Using static attributes as constant input is not something we are claiming is novel. More specifically, this method has been applied many times before in the field of machine learning (e.g. Karpathy and Fei-Fei 2014, Wen et al. 2015, Wen et al. 2016). The technique was not originally proposed by Fang et al. (2017) and their manuscript is not working on the same topic as our manuscript (rainfall-runoff modeling), we therefore do not see this as an especially appropriate reference to cite in this case.

7. L134-135. This is an interesting setup. It's worth mentioning that, from Eq 9 & 11, what was selected by the input gate were not only x_d but also h from the last step.

It is not entirely clear what the reviewer wants to suggest. If this refers to the fact that that h[t-1] is used in the forget and output gate, as well as the cell update (g[t]), then they are right. The input gate however, does not get any information of x_d[t] in our proposed EA-LSTM and neither from h[t-1]. We hope by changing the following sentence "*..while the dynamic and recurrent inputs control what information is written..*" we can resolve the confusion.

Old passage:
The static features control, through input gate (i), which parts of the LSTM are activated for any individual catchment, while the dynamic inputs control what information is written into the memory (g[t]), what is deleted (f[t]), and what of the stored information to output (o[t]) at the current time step t.

New passage:
The static features control, through input gate (i), which parts of the LSTM are activated for any individual catchment, while the dynamic and recurrent inputs control what information is written into the memory (g[t]), what is deleted (f[t]), and what of the stored information to output (o[t]) at the current time step t.

8. L158 – what happened when you used other loss functions?

We are unsure about the exact intent of this question. We used two loss functions in this manuscript and compared the results. From a hydrological modelling perspective it seems obvious that different loss functions might provide different optimization results. Designing and choosing (good/correct) objective functions is an old and important problem in hydrology. It is highly non-trivial, yet unsolved and surrounded by many discussions. However, it is also not the focus of this contribution and we therefore view the testing of more loss functions as out of scope.

9. L171 "25,000 km2" – is it really appropriate to model those with an area of 25,000 km2 the same as other smaller basins?

Although results of experiments not shown in this manuscript suggest there is no problem with doing so, in this manuscript only basins with an area smaller than 2000km² were used. As stated in L. 174 we use the same 531 basins as Newman et al. (2017): to cite their manuscript: "*We subset the complete Newman et al. (2014) basin list to remove...basins larger than 2000 km²*". That said, we agree that we missed to state this clearly in our manuscript and therefore adapt L 176 to add the following sentence "*Furthermore, out of the 671 basins, only those with an area smaller than 2000km² were kept.*"

Old passage:
We used the same 531 basins from the CAMELS data set as Newman et al. (2017). The basins are mapped in Fig. 2. These basins were chosen out of the full set because some of the basins have a large (>10 %) discrepancy between different strategies for calculating the basin area, and incorrect basin area would introduce significant uncertainty into a modeling study. The basin selection and subset is described by Newman et al. (2017).

New passage:
We used the same subselection of 531 basins from the CAMELS data set that was used by Newman et al. (2017). These basins are mapped in Fig. 2, and were chosen (by Newman et al. (2017)) out of the full set because some of the basins have a large (>10 %) discrepancy between different strategies for calculating the basin area, and incorrect basin area would introduce
significant uncertainty into a modeling study. Furthermore, only basins with a catchment area smaller than 2000 km^2 were kept.

10. L194 – "favor of"

Corrected, thank you.

Old passage:
We chose to use existing model runs so to not bias the calibration of the benchmarks to possibly favor of our own model.

New passage:
We chose to use existing model runs so to not bias the calibration of the benchmarks to possibly favor our own model.

11. L222, regarding the ensemble averaging, readers deserve to know, how big is the spread? What if you don't take the average? Sometimes the ensemble mean gets a better R2 but it misses peaks.

Yes, it is true that taking the ensemble mean will reduce variance. We've not explored more complex ensemble techniques, of which many exist. However, we see testing different ensembling strategies as out-of-scope for this paper.

12. It is unclear what "six different settings and eight different models" are.

This is explained in the preceding sentences.

- Regarding the "six different settings" L 219f "*All three model configurations were trained using the squared-error performance metrics discussed in Sect. 2.3 (MSE and NSE\*). This resulted in six different model/training configurations.*"
- Regarding the "eight different models" L 221f "*To account for stochasticity in the network initialization and in the optimization procedure (we used stochastic gradient descent), all networks were trained with n = 8 different random seeds*"

The phrase quoted by the reviewer from L. 224 (immediately following the two sentences quoted) pulls these sentences together "*In total, we trained and tested six different settings and eight different models per setting for a total of 48 different trained LSTM-type models*"

13. L261. might be useful to say you extracted gradients from the learned network after training (correct?), as some readers are unfamiliar with how this is done. However, these gradients are time-step dependent.

Indeed, we calculated the gradients w.r.t. the static inputs from the trained model, since we are interested in analyzing the robustness and feature ranking of a trained network, not of a randomly initialized one. In L. 244 we stated this fact for the model robustness "*To estimate the robustness of the trained model to uncertainty in the catchment attributes…*". We will add a similar sentence to the feature ranking to avoid possible confusion around analyzing untrained models.

Old passage:
To provide a simple estimate of the most important static features, we used the method of Morris (Morris, 1991).

New passage:
To provide a simple estimate of the most important static features of the trained model, we used the method of Morris (Morris, 1991).

14. Also, why is it called global sensitivity test? It is also local, around a origin for perturbation.

Citing Campolongo et al. (2015) from their introduction "*The Morris method is simple to understand and implement, and its results are easily interpreted. Furthermore it is economic in the sense that it requires a number of model evaluations is in the number of model factors. The method can be regarded as global as the final measure is obtained by averaging a number of local measures (the elementary effects), computed at different points of the input space.*" To clarify the result of Eq. 14 (or Eq. 15 in our case), this is not a global measure in the sense that the entire space of possible values is considered, but in the sense that more points are considered to derive the sensitivity (see Saltelli et al., 2004). This is reflected in our statement in L. 263f: "*A global sensitivity measure for each basin and each feature is then derived from taking the average absolute gradient (Saltelli et al., 2004)*"

15. L264 better say "the average of absolute gradients across all basins and all time steps", and----why absolute?

Here, we are still referring to a global sensitivity measure for each individual basin. Therefore, "*for each basin*" is correct in this sentence. The averaging across multiple basins is then applied to derive the values in Table 4 (see answer to major comment #3), after normalizing the sensitivity measures to the range (0,1) per basin. Absolute, because otherwise oscillating (positive, negative) gradients, have the potential to cancel and (erroneously) suggest that the respective feature(s) are unimportant. Furthermore, taking absolute values is the proposed

method for deriving the global sensitivity measure from these local points and is referred to as µ* in the literature (e.g. Saltelli, 2004; Campolongo et al. 2011).

16. L267-268 "represent xxx into xxx"? the sentence does not make grammar sense. please fix. This is obviously an expansion of from 27 to 256. Why would this be really necessary?

We do not see a grammatical error in this sentence. Embedding can be used as a noun, which makes the phrase "*[this] vector […] represents an embedding of xxx into yyy*" grammatically correct.

This transformation is necessary, since the resulting input gate must be a vector of 256-dimensions - the same size as the LSTM has cell states. This is basically the same as in every other gate where e.g., the 5-dimensional dynamic inputs (the 5 meteorological variables) have to be transformed into a vector of 256-dimensions for the forget and output gate and the cell update respectively.

17. Table 2. this value is indeed the highest I have seen. Good work!

Thank you.

18. L380. Why 447 basins now? What are missing?

The first sentence in Section 3.2 (L.363) explains this: "*The results in this section are calculated from 447 basins that were modeled by all benchmark models*".

It's important to reiterate that we used benchmark models that were run by the respective model development groups. We did not run our own benchmark models. This is critical because we want to give the benchmark models the highest possible chance of success - the presumption being that the respective model development groups are the most well-qualified to run their own models. Notice that this is a common strategy in model intercomparison and model benchmarking studies (e.g., Best et al., 2015)

19. L410 Unsure how this answers the question if the network just remembers. The logic is Confusing.

We think that the general results of the robustness analysis (as shown in the boxplot in Fig. 6) indeed address whether the network is simply remembering basins. If we understand the reviewer correctly, s/he is referring to pure overfitting against the static attributes and that the LSTM simply remembers all 531 catchments individually. If the LSTM simply remembers all 531 catchments individually, there would not be slow degradation in performance (as seen as increase in the variance of the boxplot over increasing level of additive noise) but rather a more drastic performance drop, when not using the exact catchment attributes for each basin. We will add a sentence that better describes this result.

Old passage:
As expected, the model performance degrades with increasing noise in the static inputs. However, the degradation does not happen abruptly but smoothly with increasing levels of noise.

New passage:
As expected, the model performance degrades with increasing noise in the static inputs. However, the degradation does not happen abruptly but smoothly with increasing levels of noise, which is an indication that the LSTM is not over-fitting on the static catchment attributes. That is, it is not remembering each basin with its set of attributes exactly, but rather learns a smooth mapping between attributes and model output.

20. L414, mean precipitation, etc --- aren' these supposed to be climatic inputs rather than static? (can we not let the network generalize it from the forcing data)?

Mean precipitation, high precipitation duration etc. are indeed climatic inputs, but also static inputs since these are aggregated values over the time series (see Addor et al. 2017). The network would only be able to derive statistics like mean precipitations internally from the time length we derive as the input for predicting a single day (here we use an input sequence length of 365 days).

21. Table 4. Echoing a major point raised above. What further conclusions can be drawn from the fact that climatic attributes take the most important positions? catchment Co-evolution theory?

Indeed, what could be inferred here? It's a good question. Certainly this type of speculation is far outside the scope of this paper. We are not prepared to speculate on climate-driven catchment co-evolution, but we suspect that the 30-year data record in CAMELS is not long enough to address this question

22. L454 "before vs. after the transformation into the embedding layer". This is a good comparison, although later there didn't seem to be much comment on this comparison

There are a few comparisons made throughout the analysis:

- L 453ff "*In all cases with cluster sizes less than 15, we see that clustering by the values of the embedding layer provides more distinct catchment clusters than when clustering by the raw catchment attributes*"
- L 462ff "*In both the k = 5 and k = 6 cluster examples, clustering by the EA-LSTM embedding layer reduced variance in the hydrological signatures by more or*

*approximately the same amount as by clustering on the raw catchment attributes. The exception to this was the hfd-mean date, which represents an annual timing process (i.e., the day of year when the catchment releases half of its annual flow). This indicates that the EA-LSTM embedding layer is largely preserving the information content about hydrological behaviors, while overall increasing distinctions between groups of similar catchments*"

- L 471ff "*Although latitude and longitude were not part of the catchment attributes vector that was used as input into the embedding layer, both the raw catchment attributes and the embedding layer clearly delineated catchments that correspond to different geographical regions within the CONUS*"

For each of the steps of the cluster analysis (silhouette plots, variance reduction and cluster results shown on the map of the USA), we actually gave a direct comparisons between the results using the embedding of the EA-LSTM or using the raw catchment attributes. We are unsure what kind of additional comments are expected from the reviewer.

23. UMAP—might be good to briefly explain what it does. Is it just PCA?

We agree that the explanation of the UMAP method could be extended in Section 2.6.3 and will update the manuscript accordingly.

New passage describing UMAP:
Finally, we reduced the dimension of the input gate embedding layer (from $R^{256}$ to $R^2$) so as to be able to visualize dominant features in the input embedding. To do this we use a dimension reduction algorithm, called UMAP (McInnes et al., 2018) for "Uniform Manifold Approximation and Projection for Dimension Reduction". UMAP is based on neighbour graphs (while e.g., principle component analysis is based on matrix factorization), and it uses ideas from topological data analysis and manifold learning techniques to guarantee that information from the high dimensional space is preserved in the reduced space. For further details we refer the reader to the original publication by McInnes et al. (2018).

24. L479 Honestly, it's not that easy to see which cluster you are talking about. could use some annotation on the plot.

This is a good idea and we will update the plot accordingly.

25. L489 I found this discussion, as a take-home message, to be somewhat superficial, and unsurprising. I'd appreciate somewhat more in-depth discussion about the hydrology.

We understand and appreciate the reviewer's perspective (more is usually better), however it's hard to see from this comment what the reviewer finds missing in our analysis. We did give quite a lot of hydrological discussion in the context of analyzing the embedding layer - does the reviewer see something in our analysis that is missing? Is there something they might hope to

learn that we didn't explore? We would love suggestions about how to improve this analysis, but just asking for more is not really an actionable suggestion.

Regarding the reviewers' suggestion that our conclusions were not surprising, I guess surprising is somewhat subjective. We were generally happy that (a) the model performed as well as it did against benchmarks, and (b) that the similarity analysis generally agreed with previous literature. This means that the model at least appears to be giving the right answers for the right reasons.

26. Figure 11 these colors do not mean anything. It is a bit confusing. Why not use a at least partially consistent color scheme?

These colors actually do mean something (and it was actually somewhat difficult to get the colors to match on the various plots). These colors present the results of several clustering analyses, and are categorical labels. Therefore, we chose to color the basins in a categorical color palette, where each color reflects one cluster class. This makes a categorical color-scheme necessary, since there is no intrinsic ordering (excluding continuous, sequential and diverging color schemes). Furthermore, we made sure that the clusters between the different subplots are more or less colored similarly, so it is easier to compare between the subplots. What else does the reviewer meant by "partially consistent color scheme"? Consistent with what? Certainly the color scheme is consistent between subplots in the figure.

27. Figure 12 better annotate axes even if they don't mean much

We decided to exclude the 2D-coordinates of the UMAP embedding because, as the reviewer suggested herself/himself, they do not mean anything. We would therefore argue that they are probably more confusing and distracting and the reader could ask what why this basin has an embedding coordinate of (4,2) and the other basin only of (0,-0.5) (both are arbitrary sets).

28. L524. I am confused why this is called regional, as the LSTM was trained with all basins over CONUS. What would constitute a model that is not regional?

Regional modeling in Hydrology has a very specific meaning. The alternative is a local model (i.e., one that is calibrated to a specific basin). The second reviewer suggested that this is potentially a universal rainfall-runoff model that could be applied to basin groups of any scale (small regions, US scale, continental scale or even globally), but our intent was to draw a connection with what is a named (and well-defined) problem in Hydrology.

29. L529-530. It either goes against a belief or it does not. Can't go "somewhat against". And, the logic here is not quite clear. This paper is not about parameter identification. The fact that the network works does not imply that parameters can be identified. First the LSTM parameters cannot be interpreted. Second, even very different parameters could give you similar predictions.

First, it's actually possible for two opinions to partially disagree or somewhat disagree, however we changed this part to: "*This result challenges the idea that runoff time series alone only contain enough information…*"

Secondly, we are not sure if we understand the comment about parameter identification:

- The technical correctness of the statement "*the LSTM parameters cannot be interpreted*" depends on one's understanding of interpretation (for a lengthy discussion on this topic we refer to Lipton, 2016). The LSTM parameters are (maybe) not one-to-one translatable into physical properties as some of the hydrological model parameters, however this criticism is no less valid for conceptual models: it might be questionable what a e.g., catchment-wide infiltration value represents.

- However, the function of each individual parameter in the trained LSTM could indeed be interpreted, to see if a certain weight e.g., thresholds to specific temperatures in the input. The huge number of parameters however, makes such work difficult. This is not really related to the point of the sentence in question, however, which is about deficiencies in hydrology models, not about the interpretability of LSTM parameters.

- We are also not sure if we understand the second point of the review in this context. Indeed, different parameters can give similar predictions and overall performances, as we have shown in the paper. However, what is the point here regarding our statement that we think traditional large-scale hydrological models can be structurally improved?

[revised manuscript text omitted]